# The protein expression profile of ACE2 in human tissues

Feria Hikmet[1],[†] iD, Loren Méar[1],[†], Åsa Edvinsson[1], Patrick Micke[1], Mathias Uhlén[2],[3] iD & Cecilia Lindskog[1],[*] iD

## Abstract

The novel SARS-coronavirus 2 (SARS-CoV-2) poses a global challenge on healthcare and society. For understanding the susceptibility for SARS-CoV-2 infection, the cell type-specific expression of the host cell surface receptor is necessary. The key protein suggested to be involved in host cell entry is angiotensin I converting enzyme 2 (ACE2). Here, we report the expression pattern of ACE2 across > 150 different cell types corresponding to all major human tissues and organs based on stringent immunohistochemical analysis. The results were compared with several datasets both on the mRNA and protein level. ACE2 expression was mainly observed in enterocytes, renal tubules, gallbladder, cardiomyocytes, male reproductive cells, placental trophoblasts, ductal cells, eye, and vasculature. In the respiratory system, the expression was limited, with no or only low expression in a subset of cells in a few individuals, observed by one antibody only. Our data constitute an important resource for further studies on SARS-CoV-2 host cell entry, in order to understand the biology of the disease and to aid in the development of effective treatments to the viral infection.

**Keywords** ACE2; immunohistochemistry; respiratory system; SARS-CoV-2; transcriptomics

**Subject Categories** Methods & Resources; Microbiology, Virology & Host Pathogen Interaction

**Mol Syst Biol. (2020) 16: e9610**

## Introduction

Coronaviruses are enveloped RNA virions, and several family members of *Coronaviridae* belong to the most prevalent causes of common cold. Two previous coronaviruses that were transmitted from animals to humans have caused severe disease: the severe acute respiratory syndrome coronavirus (SARS-CoV) and the Middle East respiratory syndrome coronavirus (MERS-CoV). The novel coronavirus SARS-coronavirus 2 (SARS-CoV-2), which shares ~80% amino acid identity with SARS-CoV, is the agent of the coronavirus disease 19 (COVID-19), the new rapidly spreading pandemic.

When coronaviruses enter the target cell, a surface unit of the spike (S) glycoprotein binds to a cellular receptor. Upon entry, cellular proteases cleave the S protein which leads to fusion of the viral and cellular membranes. SARS-CoV has previously been shown to enter the cell via the ACE2 receptor, primed by the cellular serine protease TMPRSS2 (Li *et al*, 2005; Matsuyama *et al*, 2010). Recent studies suggest that also SARS-CoV-2 employs ACE2 and cellular proteases for host cell entry, including TMPRSS2, CTSB, and CTSL (Hoffmann *et al*, 2020) and that the affinity between ACE2 and the SARS-CoV-2 S protein interaction is high enough for human transmission (Shang *et al*, 2020).

For a full understanding of the susceptibility for SARS-CoV-2 infection and the role of ACE2 in clinical manifestations of the disease, it is necessary to study the cell type-specific expression of ACE2 in human tissues, both on the mRNA and protein level. The respiratory system is of special interest due to its high susceptibility to inhaled viruses, and most viruses infecting humans use different airway epithelial cells for replication. Given the fact that COVID-19 leads to severe respiratory symptoms, in-depth characterization of both the upper and lower respiratory tract is of high priority. Expression of ACE2 in the lung or upper respiratory epithelia would suggest that these cells can serve as a reservoir for viral invasion and facilitate SARS-CoV-2 replication. It is however important to study other tissue locations expressing ACE2 that could serve as potential points for host entry, or explain the systemic clinical processes related to severe disease progression in infected individuals.

ACE2 is a carboxypeptidase negatively regulating the renin–angiotensin system (RAS), which can induce vasodilation by cleaving angiotensin II. The protein was identified in 2000 as a homolog to ACE (Tipnis *et al*, 2000) and due to the importance of ACE in cardiovascular disease, and the use of ACE inhibitors for treatment of high blood pressure and heart failure, there has been a large interest in understanding the function and expression of ACE2 in various human organs. Early studies based on multi-tissue Northern blotting and QRT-PCR have suggested moderate to high transcript expression of ACE2 in kidney, testis, heart, and the intestinal tract,

1  Rudbeck Laboratory, Department of Immunology, Genetics and Pathology, Uppsala University, Uppsala, Sweden
2  Science for Life Laboratory, School of Engineering Sciences in Chemistry, Biotechnology and Health, KTH - Royal Institute of Technology, Stockholm, Sweden
3  Department of Neuroscience, Karolinska Institutet, Stockholm, Sweden
   *Corresponding author. Tel: +46 18 471 5032; E-mail: cecilia.lindskog@igp.uu.se
   †These authors contributed equally to this work

while the expression was low or absent in lung (Tipnis *et al*, 2000; Harmer *et al*, 2002). More recently, several large-scale transcriptomics efforts based on next-generation sequencing allow for quantitative assessment across all major human tissue types (Keen & Moore, 2015; Uhlen *et al*, 2015; Yu *et al*, 2015). Complementing these datasets with single-cell RNA-seq (scRNA-seq) provides an excellent tool in studies of transcript levels expected to be found in smaller subsets of cells in complex tissue samples. Several recent studies using scRNA-seq identified ACE2 expression in specific cell types, both in the respiratory system and other organs (Lukassen *et al*, 2020; preprint: Muus *et al*, 2020; Sungnak *et al*, 2020). The spatial localization of ACE2 on the protein level across the entire human body is however still poorly understood and necessary for determining how the expression on the mRNA level can be translated to physiological and functional phenotypes.

In the present investigation, we performed a large-scale stringent immunohistochemical analysis based on two independent antibodies and provide a comprehensive summary of ACE2 *in situ* protein expression levels in > 150 different cell types corresponding to 45 different human tissues. The analysis included an in-depth characterization of human airways using large sections of nasal mucosa and bronchus, as well as an extended patient cohort of 360 normal lung samples. The results were compared with transcriptomics, scRNA-seq, Western blot, and mass spectrometry data, in order to provide a comprehensive overview of ACE2 expression across all major human tissues and organs at different levels.

# Results

## Transcriptomic profiling of ACE2

For understanding human physiology and disease, it is necessary to quantify the expression of all protein-coding genes across different organs, tissues, and cell types and study the expression in a tissue-restricted manner. Several recent large-scale transcriptomics efforts provide a framework for defining the molecular constituents of the human body. Three such body-wide initiatives include the Human Protein Atlas (HPA) consortium (Uhlen *et al*, 2015) and the genome-based tissue expression (GTEx) consortium (Keen & Moore, 2015) based on RNA-seq, and the FANTOM5 consortium (Yu *et al*, 2015) based on cap analysis gene expression (CAGE). The HPA consortium aims at characterizing the entire human proteome in tissues, organs, cells, and organelles using a combination of transcriptomics and antibody-based proteomics. In the open-access database www.proteinatlas.org, expression profiles based on both external and internal datasets are shown, together with primary data, criteria used for validation and original high-resolution images of immunohistochemically stained human tissue samples. The HPA displays mRNA expression data from HPA, GTEx, and FANTOM5 and also summarizes the three datasets as normalized expression levels (NX) across 61 tissues, organs, and blood cell types (Uhlen *et al*, 2016b, 2019). Figure 1 shows an overview of ACE2 expression based on transcriptomics. The analysis covers all major parts of the human body, corresponding to 16 different organ systems, including blood (Fig 1A). Based on the consensus of the three datasets and NX = 1.0 as detection limit, expression above the cutoff was observed in 20 different tissue types (Fig 1B). The highest

expression was observed in the intestinal tract, followed by kidney, testis, gallbladder and heart. A few tissues showed a lower expression of < 5 NX, e.g., thyroid gland and adipose tissue, while several organs had an expression level just above or below cutoff, including liver, female reproductive organs, and lung, with some variations between the three different datasets. Brain, lymphoid tissues, skin, smooth muscle, and immune cells showed a consistent lack of expression.

## Single-cell transcriptomic profiling of ACE2

In order to study the role of ACE2 in human tissues, it is necessary to analyze the expression at a cell type-specific level, since smaller subsets of cells may express the receptor and may result in expression values below detection limit when mixed with all other cells in a complex tissue sample. Single-cell RNA-seq (scRNA-seq) constitutes an excellent tool in studies of transcript levels expected to be found in smaller subsets of cells in complex tissue samples. The largest initiative aiming to create a comprehensive map of human organs based on scRNA-seq is the Human Cell Atlas consortium, which coordinates efforts from > 1,000 different institutes across > 70 countries (Regev *et al*, 2017).

Recently, several datasets presenting the expression of ACE2 in different human tissue types, including the respiratory system, have been described. In the present investigation, nine publicly available scRNA-seq datasets from ileum (Wang *et al*, 2020b), kidney (Liao *et al*, 2020), testis (Guo *et al*, 2018), lung (Reyfman *et al*, 2019; Viera Braga *et al*, 2019; Han *et al*, 2020), bronchus (Viera Braga *et al*, 2019; Lukassen *et al*, 2020), and nasal mucosa (Viera Braga *et al*, 2019) were re-analyzed (Fig 2). The results were largely consistent with the transcriptomics datasets from HPA, GTEx, and FANTOM5, confirming high expression levels in > 60% of ileal enterocytes and > 6% of renal proximal tubules. In testis, > 3% of Leydig/Sertoli cells and peritubular cells showed a high expression of ACE2. The percentage of testicular cells expressing ACE2 could however be biased due to underrepresentation of Sertoli cells in the original study (Guo *et al*, 2018).

The analysis of human lung from three different datasets suggested an enrichment in alveolar cells type 2 (AT2), although it should be noted that expression was identified only in a very small fraction of the AT2 cells (< 1%), which is in line with other recent studies (preprint: Zhao *et al*, 2020; Zou *et al*, 2020). Low expression was also observed in 2–3% and 7% of the cells in bronchus and nasal mucosa, respectively, with the highest expression observed in ciliated cells and goblet cells.

## Protein profiling of ACE2 based on immunohistochemistry

Several recent studies and datasets both on whole tissue transcriptomics and scRNA-seq have provided careful evaluation of ACE2 expression patterns on the mRNA level. For a complete understanding of the role of ACE2 in health and disease, validation on the protein level is however necessary, as the exact localization of proteins is tightly linked to protein function. Complementing the transcriptomics datasets with imaging-based proteomics allows studying proteins at their native locations in intact tissue samples. The standard method for visualizing proteins is antibody-based proteomics using immunohistochemistry, which not only gives the

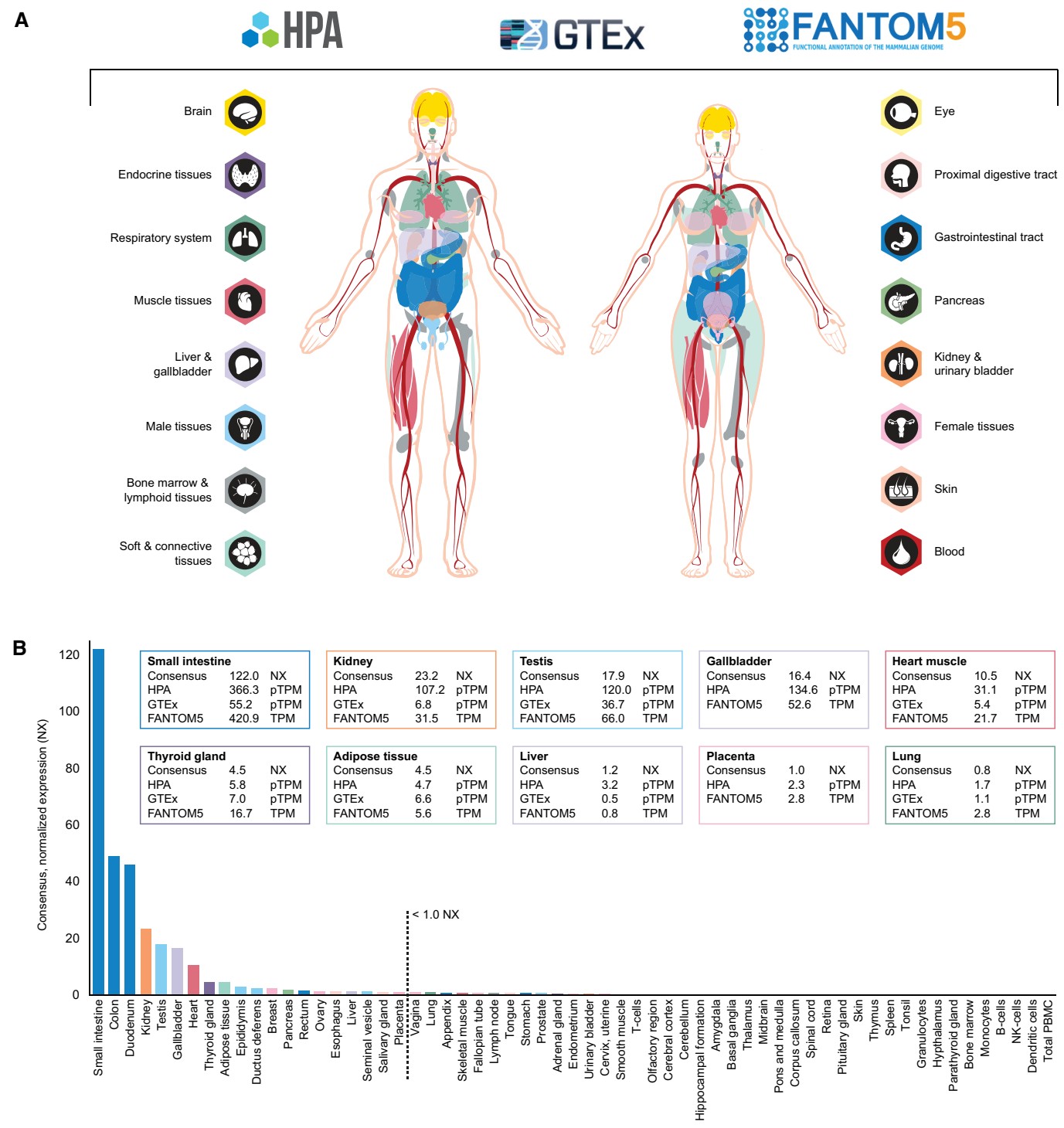

**Figure 1. ACE2 expression in human tissues based on transcriptomics.**

A  Overview of the tissues and organs analyzed based on transcriptomics by the three independent consortia Human Protein Atlas (HPA), FANTOM5, and GTEx. In total, 16 organ systems (with several tissues comprising an organ system) were used to create a consensus normalized expression (defined as the unit NX) based on the expression levels of all three datasets.

B  ACE2 gene expression summary in human tissues in 61 different tissues and cells in NX. Cutoff for what is regarded as expressed was set to 1.0 NX.

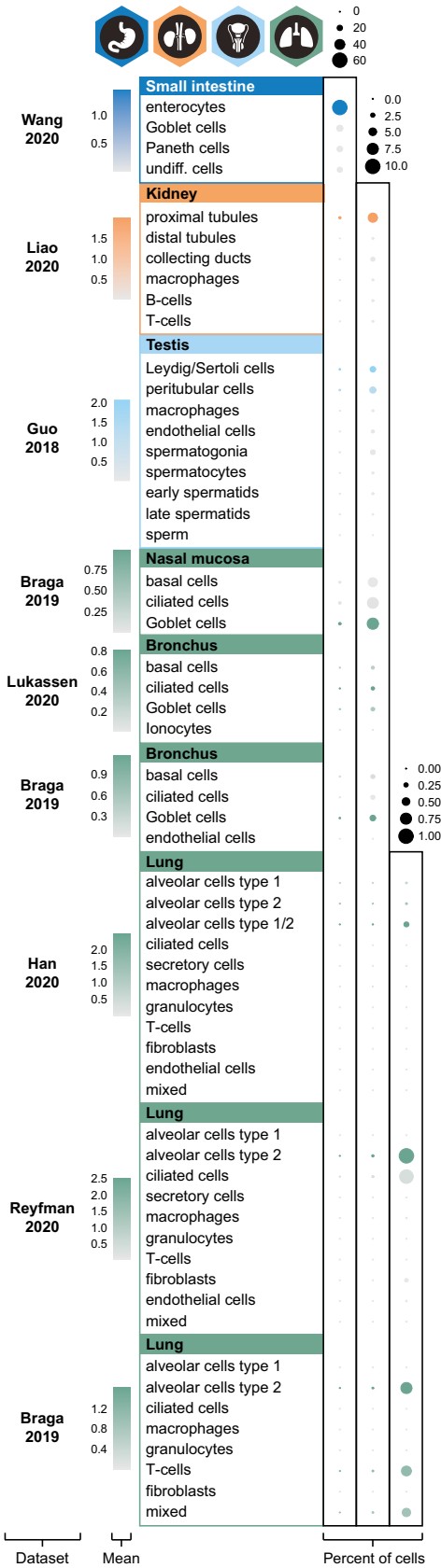

**Figure 2. ACE2 expression in human tissues based on single-cell transcriptomics.**

Dot plots summarizing the transcriptomics levels of ACE2 based on different scRNA-seq datasets and tissues. Three different scales were used, in order to be able to compare percentages of cells expressing ACE2 both between and within tissue types. The size of the dots indicates the percentage of cells expressing ACE2 in respective cell type with a maximum of 60% (left column), 10% (middle column) and 1% (right column), and the color saturation corresponds to the average expression level. Plots were generated using Seurat package in R.

possibility to define protein localization in different compartments at a single-cell and subcellular level but also provides important spatial information in the context of neighboring cells.

Standardized immunohistochemistry using two independent antibodies toward ACE2 was performed on TMAs containing up to 22 unique patient samples per tissue type from 44 different human tissues and organs (Table EV1), as well as a large section of human eye. ACE2 immunoreactivity was manually quantified in 159 different cell types, out of which 28 were positive using MAB933 (R&D Systems), and 32 were positive using HPA000288 (Atlas Antibodies). Protein expression data for all cell types using both antibodies are summarized in Table EV2. A schematic figure of ACE2-positive cell types can be observed in Fig 3A, as well as an overview of ACE2 expression patterns where at least one tissue sample showed staining with at least one antibody (Fig 3B). The results are well in line with mRNA datasets from both whole tissue transcriptomics and scRNA-seq. ACE2 immunoreactivity was observed in the intestinal tract, most abundant in duodenum and small intestine, with lower levels in stomach and large intestine (Fig 4). Consistent with high mRNA levels, distinct expression was also found in kidney, gallbladder, and testis. The expression in heart differed between the antibodies, with HPA000288 showing more prominent expression in cardiomyocytes. Several organs with lower mRNA levels, such as thyroid gland, epididymis, seminal vesicle, pancreas, liver, and placenta, showed positivity in smaller subsets of cells. High expression was also found in conjunctiva and cornea of the eye. Interestingly, small capillaries were consistently stained by both antibodies only in a few organs, including heart, pancreas, thyroid gland, parathyroid gland, and adrenal gland. Staining of ciliated cells in fallopian tube was only found for HPA000288. In general, the same tissue types showed similar expression patterns with low inter-individual variation, except for bronchus and nasal mucosa, where only one sample each out of six or eight analyzed TMA samples, respectively, showed positivity in a subset of ciliated cells for HPA000288. No immunoreactivity was observed in bronchus or nasal mucosa samples for MAB933, and none of the antibodies showed positivity in any of the lung samples. This is inconsistent with recent findings based on scRNA-seq suggesting low expression levels of ACE2 in specific cell types of lung and respiratory epithelia.

Despite mRNA expression levels above 1.0 NX, no confident expression on the protein level could be confirmed in adipose tissue, breast, ovary, esophagus or salivary gland. All other tissues with very low or absent expression based on transcriptomics were consistently negative also on the protein level, including brain, lymphoid tissues, skin, smooth muscle and immune cells (Fig EV1).

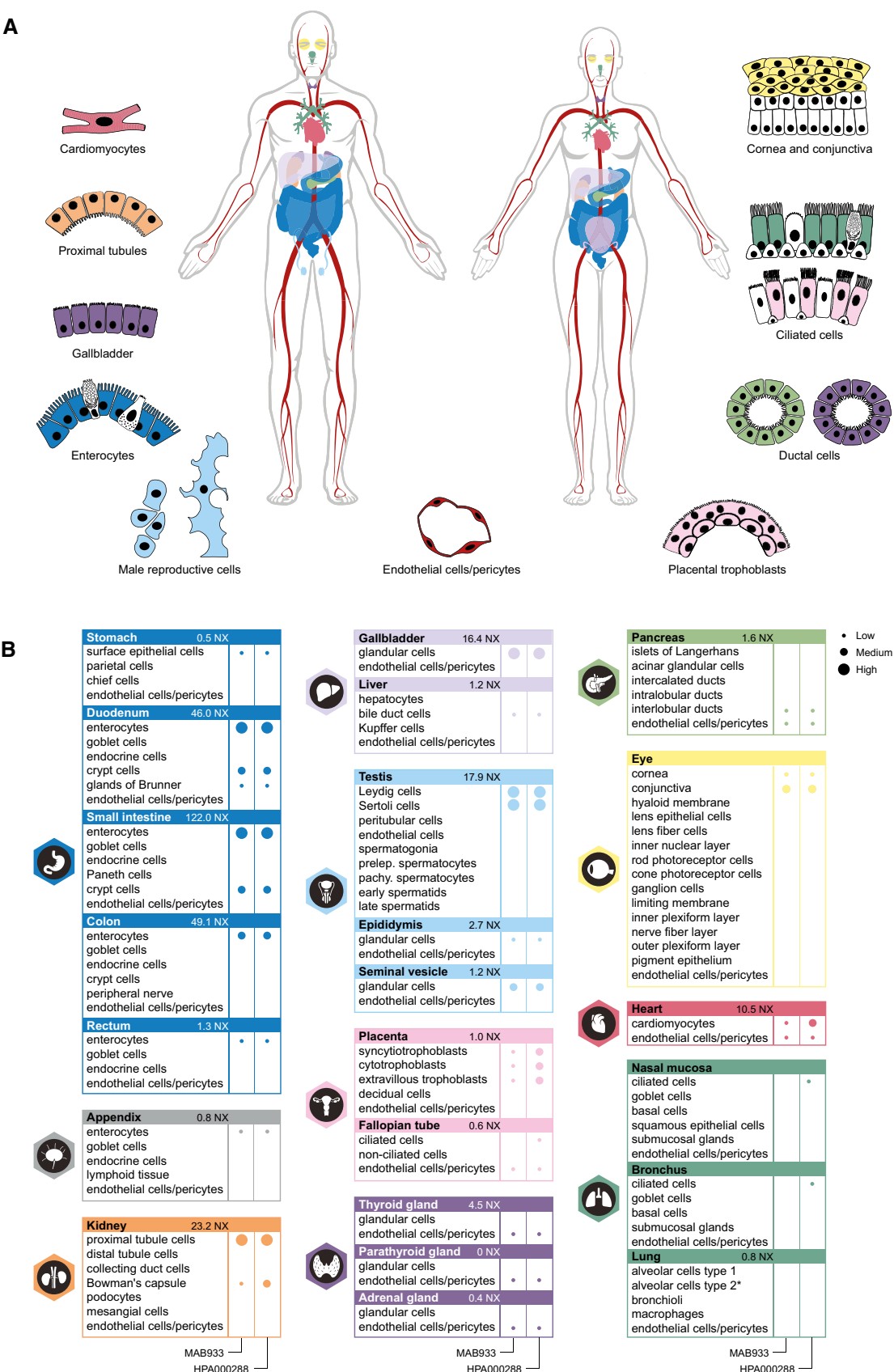

**Figure 3.**

**Figure 3. ACE2 protein expression in human tissues based on immunohistochemistry.**

A   Summary of cell types positive for ACE2 using at least one antibody. Illustrations were in part adapted from https://biorender.com/.
B   Details of cell type-specific protein expression levels based on immunohistochemistry in tissues showing distinct immunohistochemical staining in at least one cell type using at least one antibody. Left panel: MAB933 (R&D Systems). Right panel: HPA000288 (Atlas Antibodies). Dot size: level of immunohistochemical staining based on staining intensity and fraction of positive cells. Asterisk: no ACE2 protein expression was observed in the standard TMA analysis, however ACE2 protein expression in lung AT2 cells was detected in an extended lung TMA cohort.

## Immunohistochemical analysis of ACE2 in the upper and lower respiratory tract

Given the importance of the respiratory system for further understanding of SARS-CoV-2 infection and the fact that expression in these tissues could not be confidently confirmed on the protein level, we decided to perform an in-depth characterization of both the upper and lower respiratory tract in order to determine if rare expression had been missed in the initial analysis. An extended immunohistochemical analysis was performed for both antibodies on eight large sections of bronchus, 12 large sections of nasal mucosa, as well as a TMA cohort comprising histologically normal lung samples from 360 individuals (Table EV3). The results for bronchus and nasal mucosa were consistent with the initial TMA analysis, with cilia staining for the HPA000288 antibody only in a subset of cells in a few individuals, more abundant in nasal mucosa compared to bronchus (Fig 5, Table 1). In the lung cohort, only two individuals showed positivity for the HPA000288 antibody in structures most likely representing AT2 cells, while all other samples were negative (Fig 5, Table 1). The MAB933 antibody showed ACE2 expression of AT2 cells in one lung individual only, while all other samples of both the upper and lower respiratory tract were negative.

## Protein profiling of ACE2 based on Western blot and mass spectrometry

While immunohistochemistry has the advantage of determining spatial localization of proteins, the method is only semi-quantitative. Another method for determining protein levels in a tissue sample is mass spectrometry. Based on publicly available mass spectrometry datasets and protein abundance data obtained from PaxDB, ACE2 seemed to be most abundant in kidney and testis, followed by gallbladder, liver and heart (Fig 6A). No expression was observed in nasal mucosa, lung, oral mucosa or esophagus. Similar results were obtained with data from ProteomicsDB (Fig 6B), which underlined a high expression of ACE2 in kidney, rectum, ileum and testis, but no data were available for lung or upper airways. PaxDB and ProteomicsDB both take into consideration the large-scale studies based on mass spectrometry of different normal human tissues published in 2014 (Kim *et al*, 2014; Wilhelm *et al*, 2014). Data from these studies generally correlated well with mRNA expression levels from HPA, GTEx and FANTOM5, with higher expression observed in, e.g., kidney and testis, and no expression in lung. Since methods used in these studies were untargeted, it cannot be ruled out that ACE2 was expressed at lower levels in lung.

In order to determine the specificity of the ACE2 antibodies used for immunohistochemistry with another method, both antibodies were analyzed with Western blot, using lysates from lung (one male and one female), kidney, colon and tonsil (Fig 6C). Both antibodies showed a strong band corresponding to 100–130 kDa in kidney. A weaker band of slightly lower size was seen for MAB933 in colon, while no bands were detected in lung or tonsil for any of the antibodies.

# Discussion

ACE2 is suggested to be the key protein involved in SARS-CoV-2 host cell entry. Here, we present the overall expression profile of ACE2 using multiple technologies, to gain insights into the importance of this protein for the development of treatments and vaccines to combat COVID-19 and related diseases. Using an integrated omics approach, we here analyzed the spatial localization of ACE2 on the protein level in > 150 different cell types corresponding to all major human tissues and organs, and compared the expression profiles with multiple transcriptomics datasets.

In order to achieve the best estimate of protein expression across tissues and cells, stringent antibody validation strategies are needed. The International Working Group for Antibody Validation (IWGAV) formed with representatives from several major academic institutions (Uhlen *et al*, 2016a) has proposed five different pillars to be used for antibody validation to ensure reproducibility of antibody-based studies and promote stringent strategies for validation. To ensure that an antibody binds to the intended protein target, the validation must be performed in an application-specific manner, using at least one of the strategies suggested by IWGAV (Edfors

**Figure 4. Cell type-specific localization of ACE2 in human tissues based on immunohistochemistry.**

Representative images of 20 tissue types and histological structures stained on consecutive sections with immunohistochemistry using two antibodies targeting human ACE2 protein (brown), and counterstained with hematoxylin (blue). Most intense antibody staining was observed in microvilli of the intestinal tract and renal proximal tubules, in membranes of gallbladder epithelium, epididymis epithelium, testicular Sertoli cells and Leydig cells, a subset of glandular cells in seminal vesicle and cytoplasm of cardiomyocytes, with HPA000288 also staining the cardiac muscle fibers, while MAB933 only showed staining in a few cells. Distinct ACE2 staining for both antibodies was also present in cornea and conjunctiva of the eye, interlobular pancreatic ducts, as well as in placental villi, both in cytotrophoblasts, syncytiotrophoblasts, and also in extravillous trophoblasts, while placenta decidua was negative. ACE2 staining could be observed at the base of ciliated fallopian tube epithelium, however only for one of the antibodies. Note that ACE2 protein expression was less prominent in the crypts of the mucosal intestinal layer. ACE2 was also positive in endothelial cells and pericytes in several tissues, see fallopian tube, thyroid, parathyroid, adrenal gland, pancreas, and heart. Scale bar = 50 µm. Scale bar in dashed squares = 10 µm (Brunner = Brunner's gland, EVT = extravillous trophoblasts, endo = endothelial cells).

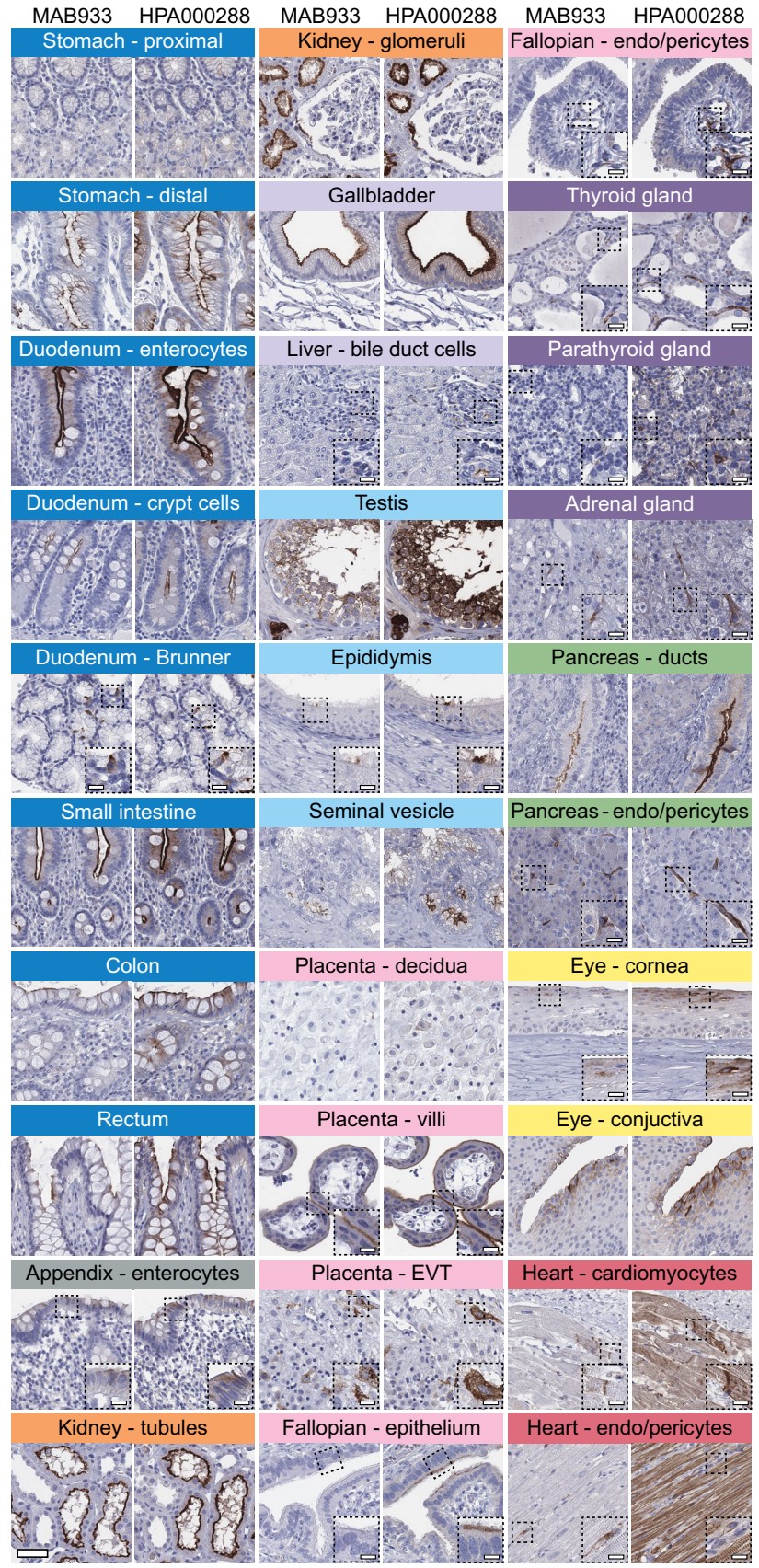

**Figure 4.**

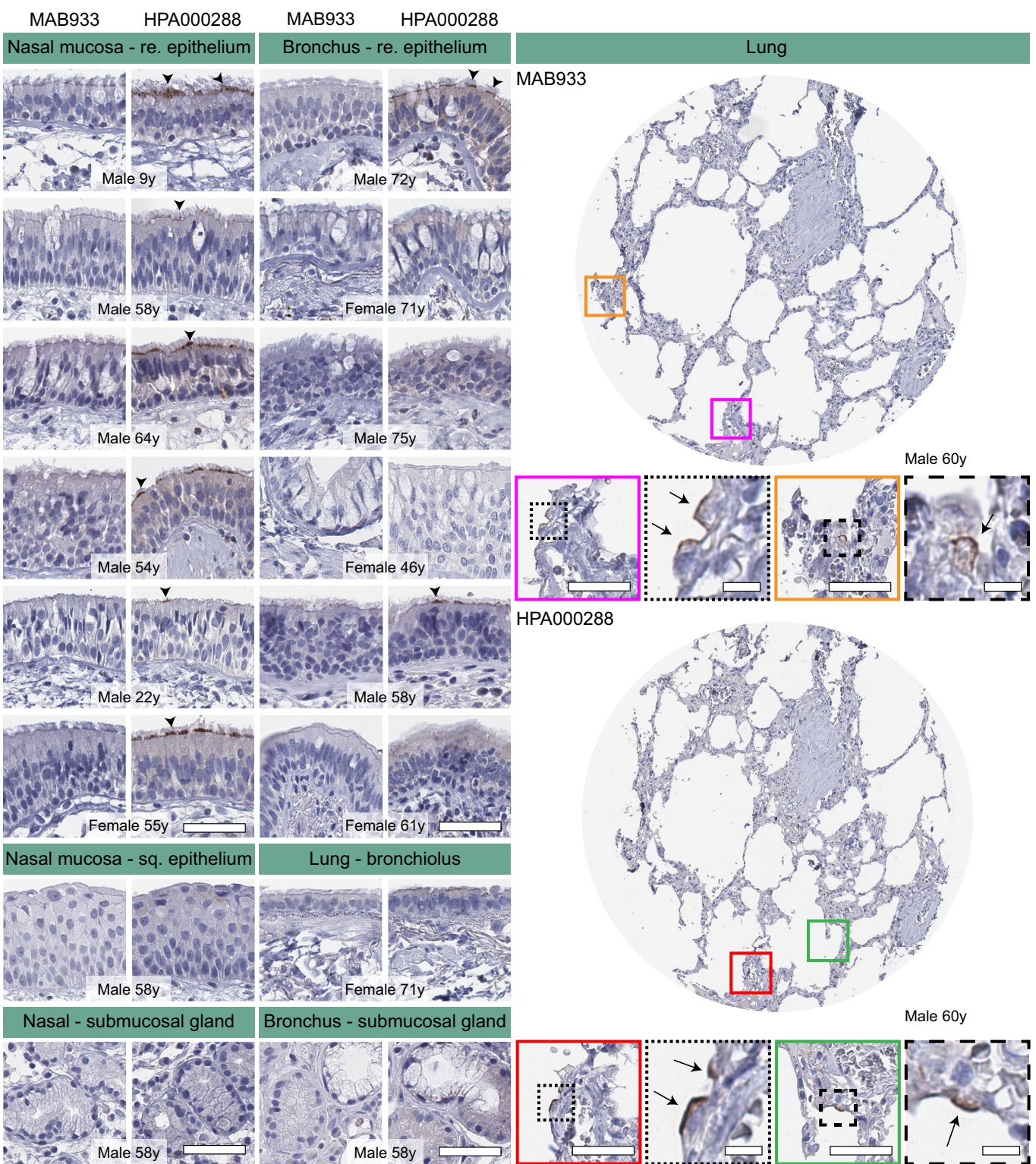

**Figure 5. Cell type-specific localization of ACE2 in human respiratory system based on immunohistochemistry.**

Representative images of human respiratory tissues, all stained on consecutive sections, with immunohistochemistry using two antibodies targeting human ACE2 protein (brown), and counterstained with hematoxylin (blue). Respiratory tissues were composed of different structures in nasal mucosa, bronchus, smaller bronchioles, and lung parenchyma. ACE2 staining could be observed at the base of ciliated cells in both nasal mucosa and bronchial epithelium (arrowheads). Rare ACE2 staining was present in a few alveolar cells (arrows). No staining was observed in nasal squamous epithelium, bronchioles, or submucosal glands in either tissue. Gender and age are shown for all individuals. Red and green colored squares mark the positions in the TMA cores shown as magnifications. Scale bar = 50 μm. Scale bar for images in dashed squares = 10 μm (re = respiratory; sq = squamous).

**Table 1. Summary of ACE2 positive and negative cases in extended normal lung TMA cohort and whole slide tissue sections of nasal mucosa and bronchus.**

| Tissue | ACE2 IHC | MAB933 | | HPA000288 | |
| --- | --- | --- | --- | --- | --- |
| | | Female | Male | Female | Male |
| Nasal mucosa (*n* = 12) | Positive | 0 | 0 | 1 | 5 |
| | Negative | 1 | 11 | 0 | 6 |
| Bronchus (*n* = 8) | Positive | 0 | 0 | 0 | 2 |
| | Negative | 4 | 4 | 4 | 2 |
| Lung (*n* = 360) | Positive | 0 | 1 | 1 | 1 |
| | Negative | 183 | 176 | 182 | 176 |

The number of positive or negative cases is shown for each antibody and gender of each tissue.

*et al*, 2018). For immunohistochemistry, two different strategies for antibody validation can be used: orthogonal strategy, based on comparison of protein expression levels using an antibody-independent method analyzing the expression levels of the same target across tissues expressing the target protein at different levels (1); or independent antibody strategy, defined as a similar expression pattern observed by an independent antibody targeting the same protein (2). In the present investigation, both antibodies used for ACE2 immunohistochemistry passed validation criteria suggested by IWGAV. The overall protein expression patterns showed a high correlation with mRNA expression levels from three different datasets (HPA, GTEx and FANTOM5), thereby validating the antibodies using an orthogonal method. Protein expression levels based on immunohistochemistry were also highly consistent with tissue levels observed by mass spectrometry-based proteomics. In addition, the protein expression patterns were in general similar between the two different antibodies that were raised toward partly overlapping epitopes.

We could confidently present the cell type-specific localization of ACE2 across several tissues with consistent high expression levels both on the mRNA and protein level, including the intestinal tract, kidney, gallbladder, heart, and testis. Several recent studies have highlighted COVID-19 related symptoms in organs expressing high levels of ACE2, including gastrointestinal symptoms (Guan *et al*, 2020), renal failure (preprint: Cheng *et al*, 2020; preprint: Wang *et al*, 2020a), cardiac injury (Huang *et al*, 2020; preprint: Hui *et al*, 2020a; Zheng *et al*, 2020), and effects on male gonadal function (preprint: Ma *et al*, 2020). SARS-CoV-2 has also been detected in stool (Xu *et al*, 2020), urine (Peng *et al*, 2020), and semen (Li *et al*, 2020) samples. Recently, it was shown that the virus can productively infect human gut enterocytes, highlighting the need to further study virus shedding in the gastrointestinal tract and the possibility of fecal–oral transmission. While the transcriptomics datasets from HPA, GTEx, and FANTOM5 did not include whole samples of human eye, both antibodies showed distinct ACE2 expression in superficial layers of cornea and conjunctiva. Interestingly, *ex vivo* cultures of human conjunctiva have shown high tropism for SARS-CoV-2, and a recent study also suggested transcript expression of ACE2 in corneal epithelium (Hui *et al*, 2020b; preprint: Sun *et al*, 2020).

The use of strict IWGAV recommendations has major advantages in confirming expression patterns that consistently show a high level of expression based on orthogonal methods and more than one antibody. It is also useful for confidently identifying tissues that

are expected to lack expression and can serve as negative controls. In the present investigation, no dataset showed ACE2 expression in the brain, lymphoid tissues, skin, smooth muscle, and immune cells. However, it is important to point out that identification of mRNA transcripts or proteins expressed at very low levels can be inconclusive. Several organs that based on transcriptomics showed low ACE2 mRNA levels were here found to be positive in smaller structures in the immunohistochemical analysis. Such structures include subsets of epithelial cells in epididymis or seminal vesicle, or capillaries that correspond to a small fraction of the total amount of cells in a certain sample. Such staining patterns that are represented by both antibodies are highly interesting for further studies. In this context, it is worth noting that expression of ACE2 has recently been described in the vasculature, suggesting a high expression of ACE2 in microvascular pericytes. One could speculate that the expression in the vasculature in heart or other organs may explain clinical manifestations of the SARS-CoV-2, such as hyperinflammation, coagulation, vascular dysfunction, and troponin leakage, resulting in a higher risk of thromboembolism and critically ill conditions (Chen *et al*, 2020a; preprint: He *et al*, 2020; preprint: Muus *et al*, 2020). Further colocalization studies using high-resolution microscopy are needed in order to confirm if the vascular expression identified here is in fact endothelial cells or pericytes, but the consistent expression in smaller capillaries of several organs, especially in heart and the endocrine system, highlights the importance to explore the role of the vasculature in SARS-CoV-2 infection. Both antibodies also showed distinct expression in placental trophoblasts, most prominent with one of the antibodies. The high ACE2 protein expression in syncytiotrophoblasts is consistent with recent studies on SARS-CoV-2 invasion of human placenta, where the virus was localized predominantly to syncytiotrophoblasts (Hosier *et al*, 2020). This highlights the importance to further explore the possibility of transmission of SARS-CoV-2 from mother to fetus.

The original report of ACE2 protein expression showed positivity in most tissue and cell types examined, including AT2 cells (Hamming *et al*, 2004). However, only one antibody was used in this study and the antibody showed strong staining in several organs that lack mRNA expression according to the data from HPA, GTEx, and FANTOM5 presented here. The antibody thus does not meet the criteria for enhanced validation proposed by the International Working Group for Antibody Validation (IWGAV). Several other studies are also inconclusive. As an example, the MAB933

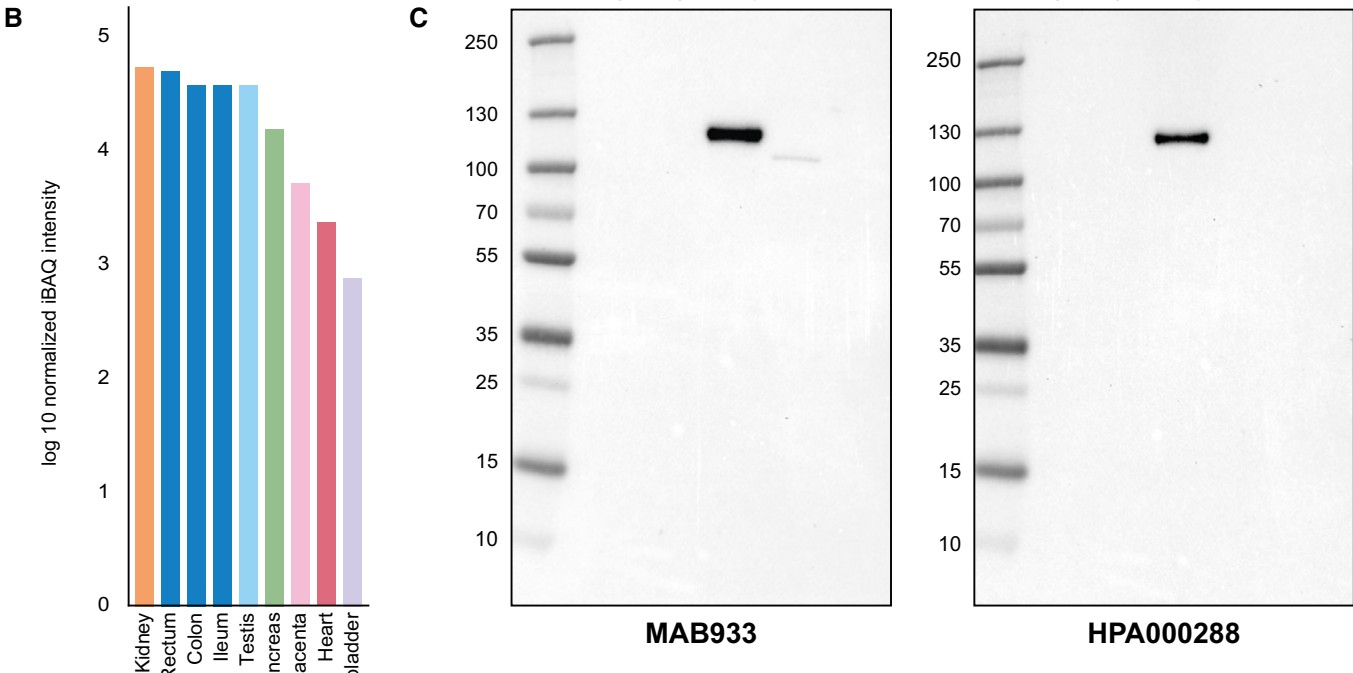

**Figure 6.**

**Figure 6. ACE2 expression in human tissues based on mass spectrometry and Western blot.**

A ACE2 protein abundance in different human tissues based on various studies processed by PaxDB, with expression levels presented as parts per million (ppm). The "Integrated" dataset corresponds to PaxDB estimation of average expression.

B ACE2 protein abundance in different human tissues from ProteomicsDB, with the median expression presented in log10(iBAQ).

C Western blot of ACE2 using five different human tissue lysates. For lung, two different lysates were used: one male (left) and one female (right).

antibody from R&D Systems that was used here was in an earlier report observed in primary cultures of human airway epithelia (Jia *et al*, 2005), while a more recent study on many individuals using the same antibody found very limited expression in the respiratory system (preprint: Aguiar *et al*, 2020). The latter study confirms the results shown in the present investigation using this antibody. Other studies showed different results between antibodies, or did not study the protein expression across many individuals in comparison with levels observed in other tissues (preprint: Lee *et al*, 2020; preprint: Muus *et al*, 2020).

While the two antibodies used in the present study showed consistent staining patterns in most tissues and cell types, only one antibody showed positivity in ciliated cells of fallopian tube and respiratory epithelium. Several recent studies based on scRNA-seq analysis of the respiratory system point at highest expression in ciliated cells, suggesting that these cells could serve as a main route of infection for SARS-CoV-2 (preprint: Muus *et al*, 2020; Sungnak *et al*, 2020). This is underpinned by the fact that swabs from the throat and nose often contain high levels of SARS-CoV-2. Novel data from experimental infection of human bronchial epithelial cells (HBECs) also highlight ciliated cells as the major target of infection (preprint: Ravindra *et al*, 2020). The staining in ciliated cells observed by one antibody in the present investigation was mainly localized to ciliary rootlets in a subset of cells in nasal mucosa from a few individuals and in only one bronchus sample. Since the staining of ACE2 was mainly observed in nasal epithelium, it would be of interest to study the difference in ACE2 expression between epithelial cells situated in the upper nasal and lower bronchial respiratory tract. In a recent study by the Human Cell Atlas Lung Biological Network and co-workers (Ziegler *et al*, 2020), it could be established that an interferon stimulation in fact leads to overexpression of ACE2 in bronchial cells, among other cell types. Based on scRNA-seq of human lung, low ACE2 expression is mainly found in a small subset of AT2 cells, often in < 1% of the cells. Here, ACE2 positivity in a few cells, most likely representing AT2 cells, was observed in two samples of a patient cohort consisting of normal lung parenchyma from 360 patients, while all other individuals were completely negative.

ACE2 protein expression based on other methods for protein detection, such as mass spectrometry and Western blot, was mainly consistent with the immunohistochemistry-based analysis. It should however be noted that these methods represent low-sensitivity approaches not well suited in the search for low abundant proteins. Antibody-based proteomics using immunohistochemistry has the advantage of spatial localization of proteins in the context of neighboring cells and is therefore useful for the detection of staining patterns observed only in small subsets of cells. The method however requires careful optimization, as even specific antibodies that have been shown to target the correct protein may generate additional unspecific off-target binding depending on antigen retrieval method, dilution or detection method (O'Hurley *et al*, 2014). While immunohistochemistry is a comparably more sensitive

approach than mass spectrometry and Western blot, it may however fail to show expression of proteins detected at very low levels, as they may fall under detection limit. Immunohistochemistry on formalin-fixed tissue samples involves cross-linking that may mask access of the antibody to certain epitopes, or antibody binding may be hindered by glycosylation. In the present investigation, two antibodies both targeting the extracellular domain of ACE2 were used for immunohistochemical analysis, and only one of the antibodies showed positivity in ciliated cells. Further studies using other antibodies with non-overlapping epitopes including antibodies targeting the cytoplasmic domain are needed to confirm these findings.

Analysis of ACE2 mRNA transcripts based on spatial transcriptomics or *in situ* hybridization would also add important information on ACE2 expression in a tissue context that can be compared with results from antibody-based proteomics. Since transcript levels largely can be considered as a proxy for protein expression levels, transcriptomics analyses constitute attractive approaches for quantitative measurements across different tissues and cell types. High-throughput mRNA sequencing based on RNA-seq of whole tissue samples is a robust and sensitive method that has low technical variation, thereby valuable for comparisons between organs and identification of genes elevated in certain organs. Bulk RNA-seq however reflects the average gene expression across thousands of cells in complex tissue samples, making the method less suitable for answering questions on cell-to-cell variations in gene expression levels. With the introduction of scRNA-seq, another level of specificity is added, revolutionizing transcriptomics studies allowing for dissecting gene expression at single-cell resolution. This new method however leads to noisier and more variable data compared to bulk RNA-seq (Chen *et al*, 2019). Technical limitations related to the different techniques used for tissue dissociation result in a lower amount of starting material for sequencing. The proportions of different cell types analyzed within a tissue may be biased and not reflect the true biological proportions, due to some cell types being less tolerant for dissociation. It may also lead to the "drop out" phenomenon, where a gene showing high expression levels in a certain cell lacks expression in other cells corresponding to the same cell type (Haque *et al*, 2017). Other limitations include challenges related to interpretation and analysis of the complex datasets, involving quality control for removing low-quality scRNA-seq data, normalization and manual annotation of cell clusters. It is also important to note that scRNA-seq data generated by different methods or platforms may lead to batch effects. To be able to compare expression data based on scRNA-seq datasets from different sources across diverse cell and tissue types in a consistent way, the global Human Cell Atlas effort promotes standardization of methods and strategies used for both scRNA-seq protocols and data analysis (Ding *et al*, 2020; Mereu *et al*, 2020). Further advances in this emerging field are likely to lead to more refined maps on the single-cell transcriptomes of different human tissues and organs in health and disease. As various methods for mRNA and protein detection have different advantages and disadvantages, an

integrated omics approach combining data across different platforms is necessary in order to obtain a complete overview of ACE2 expression across the human body.

Here, we observed none or only very low levels of ACE2 protein in normal respiratory system. Further studies are needed to investigate if the very limited staining shown in ciliated cells and AT2 cells represents a true expression of ACE2 and whether this would be enough for SARS-CoV-2 host cell entry. It also remains to be elucidated if alternate receptors may be involved to facilitate the initial infection as has been suggested by a few other groups (Ibrahim et al, 2020; Wang et al, 2020c), or the possibilities that SARS-CoV-2 enters the host via non-receptor-dependent infection. It should also be noted that the prerequisites of the interplay between SARS-CoV-2 attachment pattern to human cell surface receptors and COVID-19 disease progression are multifactorial, including not only the binding of the virus to the target cells but also other factors such as the host's immune system and the total virus load. It has been suggested that the initial host immune response to SARS-CoV-2 may trigger an interferon-driven upregulation of ACE2, leading to an increase in the number of cells in respiratory epithelia susceptible for SARS-CoV-2 infection, which could potentially explain the possibility of SARS-CoV-2 to enter via ACE2 receptors in the respiratory tract despite low ACE2 expression under normal conditions (preprint: Chua et al, 2020; Ziegler et al, 2020). Upregulation of ACE2 in respiratory epithelia of COVID-19 patients on the protein level has however not yet been shown.

While it is evident that COVID-19 leads to significant respiratory symptoms, the exact pathophysiological mechanisms of human transmission and infection are still unclear. Recent clinical descriptions of severe cases point at an inflammatory reaction both systemically and in the lungs leading to acute respiratory disease syndrome (ARDS) and in some cases death (Chen et al, 2020b; Huang et al, 2020). It however remains to be elucidated if the symptoms are driven by local virus infection of the cells in the respiratory tract or caused by secondary effects. To increase the understanding of the role of ACE2 in SARS-CoV-2 infection, we here provide an extensive and comprehensive overview of ACE2 expression across multiple datasets both on the mRNA and protein level. The results confirm expression of ACE2 in cell types previously suggested to harbor ACE2, but also adds novel insights into ACE2 expression in tissues and cell types with no previous or only limited evidence of expression. The data therefore constitute a resource for further studies on SARS-CoV-2 host cell entry with strong evidence that ACE2 expression in the respiratory tract is limited, with none or low levels of ACE2 expression in lung and respiratory epithelia. Studies using samples from COVID-19 patients are urgently needed to confirm the histological colocalization of the virus with ACE2 and other SARS-CoV-2-related proteins to gain a complete understanding of COVID-19 disease progression and the viral pathogenesis of SARS-CoV-2.

# Materials and Methods

### Data sources

RNA expression data from HPA (Uhlen et al, 2015), GTEx (Keen & Moore, 2015), and FANTOM5 (Yu et al, 2015) as well as the normalized RNA expression dataset were retrieved from the HPA database

(https://v19.proteinatlas.org). Consensus transcript expression levels were obtained through a normalization pipeline, as described previously (Uhlen et al, 2019). Protein levels based on immunohistochemistry correspond to staining intensity based on manual annotation, as described below. The lung scRNA-seq data (gene raw counts) were downloaded from the Gene Expression Omnibus (GEO) (https://www.ncbi.nlm.nih.gov/geo/) database under the series numbers: GSE130148 (Viera Braga et al, 2019), GSE134355 (Han et al, 2020) (samples: GSM4008628, GSM4008629, GSM4008630, GSM4008631, GSM4008632, and GSM4008633) and GSE122960 (Reyfman et al, 2019) (samples: GSM3489182, GSM3489185, GSM3489187, GSM3489189, GSM3489191, GSM3489193, GSM3489195, and GSM3489197). The same applies to testis, kidney, and small intestine datasets, obtained from the following accession numbers: testis GSE112013 (Guo et al, 2018), kidney GSE131685 (Liao et al, 2020), and small intestine GSE125970 (Wang et al, 2020b). Other datasets were retrieved from https://www.covid19cellatlas.org/ (Sungnak et al, 2020) including nasal brushes (Lukassen et al) and bronchi (Viera Braga et al, 2019; Lukassen et al, 2020).

ACE2 mass spectrometry-based expression among different tissues was assessed by retrieving data in two proteomics databases: Protein abundance database (PaxDB) (Wang et al, 2015) and ProteomicsDB (Schmidt et al, 2018). ACE2 protein abundances from different tissues were downloaded from PaxDB, and these values are based on MS publicly available data including large-scale studies of different normal human tissues (Kim et al, 2014; Wilhelm et al, 2014). In this database, quantification data from different datasets were processed in order to be unified, and then, if more than one value was available, integrated by weighted averaging in order to provide an estimated dataset (Wang et al, 2015). The results, presented in PPM (part per million), provide an estimation of protein abundance relative to the other proteins of the tissue. ProteomicsDB facilitated exploration of protein expression patterns, comparisons across datasets, and gave an average intensity across the human body. Protein abundance data for ACE2 were obtained from the expression tab of ProteomicsDB and presented as a normalized intensity-based absolute quantification (iBAQ) value.

### Analysis of scRNA-seq data

Data analysis including filtering, normalization, and clustering was performed using Seurat V3.0 (https://satijalab.org/seurat/9) (Butler et al, 2018) in R (CRAN). The same workflow (including normalization, scaling, and clustering of cells) was performed for each dataset. Cells were removed when they had < 500 genes and > 20% reads mapped to mitochondrial expression genome. The gene expression data were normalized using Seurat default settings, and cell clustering was based on the 5,000 most highly variable genes. All scatter plots were obtained using the UMAP method. Cluster identity was manually assigned based on well-known cell type markers and named based on the main cell type in each cluster. Seurat also allowed an intuitive visualization of ACE2 expression among the cell types thanks to the DotPlot function.

### Human tissues and sample preparation

Human tissues samples for analysis of mRNA and protein expression in the HPA datasets were collected and handled in

accordance with Swedish laws and regulation. Tissues were obtained from the Clinical Pathology department, Uppsala University Hospital, Sweden, and collected within the Uppsala Biobank organization. All samples were anonymized for personal identity by following the approval and advisory report from the Uppsala Ethical Review Board (Ref # 2002-577, 2005-388, 2007-159, 2011-473). Informed consent was obtained from all subjects in the study. The RNA extraction and RNA-seq procedure have been described previously (Uhlen *et al*, 2015). For immunohistochemical analysis, formalin-fixed, paraffin-embedded (FFPE) tissue blocks from the pathology archives were selected based on normal histology using a hematoxylin–eosin-stained tissue section for evaluation. In short, representative 1 mm diameter cores were sampled from FFPE blocks and assembled into tissue microarrays (TMAs) of normal tissue samples. Several different arrays were analyzed for each antibody, corresponding to between four and 22 unique individuals per tissue type (Table EV1). In addition to the standard TMA analysis, both antibodies were analyzed on large sections from eye, bronchus and nasal mucosa, as well as an extended cohort comprising 360 normal lung samples (Table EV3).

### Extended lung cohort

Normal lung tissue cores of 1 mm in diameter were sampled from FFPE tissue blocks of 360 patients diagnosed with lung cancer and assembled into TMAs. Only distant areas of non-malignant lung parenchyma were sampled. The cohort has been described previously (Djureinovic *et al*, 2016) and is based on consecutive patients with non-small-cell lung cancer that underwent surgical resection at Uppsala University Hospital, Uppsala, Sweden, between the years 2006 and 2010, and approved by the Uppsala Ethical Review Board (Ref# 2013/040).

### Immunohistochemistry

Immunohistochemical staining and high-resolution digitization of stained TMA slides were performed essentially as previously described (Kampf *et al*, 2012). TMA blocks were cut in 4 μm sections using a waterfall microtome (Microm H355S, Thermo Fisher Scientific, Freemont, CA), placed on SuperFrost Plus slides (Thermo Fisher Scientific, Freemont, CA), dried overnight at room temperature (RT), and then baked at 50°C for at least 12 h. Automated immunohistochemistry was performed by using Lab Vision Autostainer 480S Module (Thermo Fisher Scientific, Freemont, CA), as described in detail previously (Kampf *et al*, 2012). Primary antibodies toward human ACE2 were the polyclonal rabbit IgG antibody HPA000288, RRID:AB_1078160, (Atlas Antibodies AB, Bromma, Sweden) and monoclonal mouse IgG antibody MAB933, RRID:AB_2223153, (R&D Systems, Minneapolis, MN). The antibodies were diluted and optimized based on IWGAV criteria for antibody validation (Uhlen *et al*, 2016a). Protocol optimization was performed on a test TMA containing 20 different normal tissues. The stained slides were digitized with ScanScope AT2 (Leica Aperio, Vista, CA) using a 20× objective. All tissue samples were manually annotated by two independent observers (FH and CL) in 159 different cell types. Annotation parameters included staining intensity and quantity of stained cells, determined using a 4-

graded scale based on standardized HPA workflow (Uhlen *et al*, 2015): 0 = Not detected (negative, or weak staining in < 25% of the cells); 1 = Low (weak staining in ≥ 25% of the cells, or moderate staining in < 25% of the cells); 2 = Medium (moderate staining in ≥ 25% of the cells, or strong staining in < 25% of the cells); or 3 = High (strong staining in ≥ 25% of the cells). A consensus classification per cell type and antibody was determined taking all individual samples into consideration, and difficult cases were discussed with a third independent observer—a certified pathologist (PM).

### Western blot

Protein extracts were generated from fresh frozen lung (male/female), kidney, colon, and tonsil using a ProteoExtract Complete Mammalian Proteome Extraction Kit (cat#539779, Merck Millipore, Darmstadt, Germany), following tissue homogenization with a Precellys 24 (Bertin Instruments, France). Lysates were measured for protein concentration using a Non-Interfering Protein Assay kit (cat#488250, Merck Millipore, Darmstadt, Germany) at 480 nm and stored in −70°C.

Based on the protein concentration, the lysates were mixed with MilliQ $H_2O$ and 3xTCEP sample buffer (9 μl 3xTCEP sample buffer per 21 μl (protein lysate + MilliQ $H_2O$)). Ingredients for 1 ml 3xTCEP sample buffer were as follows: 0.187 ml 0.5 M Tris-HCl pH 6.8, 0.436 ml 87% glycerol, 0.03 ml 1% bromophenol blue, 0.3 ml 10% SDS, 0.03 ml 0.5 M TCEP pH 7, and 0.017 ml MilliQ $H_2O$. The mixed samples were heated for 5 min at 95°C, cooled down, and vortexed before loading to gel (4–20% Criterion TGX Precast Midi Protein Gel, Bio-Rad Laboratories, USA). Total amount of protein loaded per well was 15 μg. Gel was run in 1× Tris/Glycine/SDS running buffer (Bio-Rad Laboratories, USA) for 45 min at 200V. PageRuler Plus Prestained Protein Ladder (#26619 Thermo Fisher Scientific, Freemont, CA) was applied with the following molecular sizes: 250, 130, 100, 70, 55, 35, 25, 15, and 10 kDa.

For protein transfer, a Trans-Blot Turbo Transfer System (Bio-Rad Laboratories, USA) was used. The gel was placed on a PVDF membrane from a transfer pack (Bio-Rad Laboratories, USA), and the Trans-Blot Turbo Transfer System was run for 9 min at a high molecular weight setting. After the run, the membranes were dried and stored dark for further downstream testing. One membrane per batch was stained with Ponceau S (Sigma-Aldrich, Darmstadt, Germany) for 5 min to control proper protein transfer. The same membrane was then rinsed and used as negative control and run in the same manner as the other membranes except for the primary antibody step. Equilibration of the dry PVDF membrane was done in 95% EtOH, followed by rinsing with Tris-buffered saline with Tween 20 (TBST) (0.05% Tween) and a blocking step with TBST (0.5% Tween 20) + 5% dry milk in for 45 min. The membranes were incubated with two different primary antibodies toward ACE2: HPA000288 (Atlas Antibodies AB) and MAB933 (R&D Systems). HRP-conjugated antibodies were used as secondary antibodies (Swine anti-rabbit 1:3,000, Goat anti-mouse 1:5,000, Dako, Glostrup, Denmark). Membranes were developed with Clarity Western ECL Substrate 1:1 (Bio-Rad Laboratories, USA), and chemiluminescence signal was detected with a ChemiDoc Touch camera (Bio-Rad Laboratories, USA).

## Data availability

High-resolution images corresponding to immunohistochemically stained TMA cores (using both antibodies) from 44 different tissue types, as well as the lung cohort of 360 individuals and whole slide images of bronchus, nasal mucosa and eye tissue are available in the BioStudies repository (https://www.ebi.ac.uk/biostudies) under the accession S-BSST421. The normalized consensus transcript expression levels based on transcriptomics data from HPA, GTEx, and FANTOM5 are readily available under the download page in the latest version 19.3 of the Human Protein Atlas (https://v19.proteinatlas.org).

**Expanded View** for this article is available online.

## Acknowledgements

The project was funded by the Knut and Alice Wallenberg Foundation. Pathologists and staff at the Department of Clinical Pathology, Uppsala University Hospital, are acknowledged for providing the tissues used for RNA-seq and immunohistochemistry. The authors would also like to thank all staff of the Human Protein Atlas for their work.

## Author contributions

CL conceived and coordinated the study. FH, LM, and ÅE performed experiments, and all authors analyzed and interpreted data. PM provided clinical samples and pathology expertise. CL wrote the manuscript, with contributions by MU, FH, LM, and ÅE. CL, FH, and LM assembled the final figures. All authors commented on and agreed on the presentation.

## Conflict of interest

The authors declare that they have no conflict of interest.

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
