## [Review Process File · Molecular Systems Biology]

The protein expression profile of ACE2 in human tissues

Feria Hikmet, Loren Méar, Åsa Edvinsson, Patrick Micke, Mathias Uhlen, and Cecilia Lindskog
DOI: [10.15252/msb.20209610](https://doi.org/10.15252/msb.20209610)

Corresponding author(s): Cecilia Lindskog (cecilia.lindskog@igp.uu.se)

Review Timeline:

Submission Date:	3rd Apr 20
Editorial Decision:	27th Apr 20
Revision Received:	20th May 20
Editorial Decision:	24th Jun 20
Revision Received:	1st Jul 20
Accepted:	2nd Jul 20

Editor: Maria Polychronidou

Transaction Report:

27th Apr 2020

Manuscript Number: MSB-20-9610

Title: The protein expression profile of ACE2 in human tissues

Thank you again for submitting your work to Molecular Systems Biology. We have now heard back from the three referees who agreed to evaluate your study. Overall, the reviewers think that the presented findings seem interesting and are relevant in the context of understanding the role of ACE2 in infection by covid-19. They mention however that as it stands the study seems somewhat preliminary and they raise a series of concerns, which we would ask you to address in a major revision.

As you will see below, the reviewers point out that additional analyses are needed and a more careful and balanced interpretation of the data is warranted. The reviewers make constructive suggestions on how to improve the study. Without repeating all the concerns listed below, some of the more fundamental issues are the following:

- Reviewer #1 raises a very good point, which refers to the potentially important insights that could be derived by analyzing data from covid-19 patient samples. In case you have access to such material we would strongly encourage you to perform these analyses, as they would significantly enhance the conclusiveness and impact of the study and could address questions including the effect of immune responses on ACE2 levels, the link to the expression of the protease TMPRSS2 etc.
- Reviewer #1 mentions that further proteomics resources need to be considered in the analyses.
- A more balanced interpretation and discussion of alternative explanations and hypotheses is warranted to provide a more informative view on what the observed low ACE2 protein levels might mean for covid-19 infection.
- As reviewers #2 and #3 recommend, a more thorough comparison and discussion of RNA-seq and scRNA-seq data needs to be included.

All other issues raised by the reviewers should be convincingly addressed. Please let me know in case you would like to discuss any of the issues raised.

On a more editorial level, we would ask you to address the following.

REFEREE REPORTS

Reviewer #1:

Review of MSB-20-9610

In the present manuscript, Hikmet et al analyse the (protein) expression of ACE2 using antibodies in 78 different human cell types. The study is motivated by the current SARS-CoV-2 pandemic and the presumed essential role of ACE2 in the infection process. Given the perceived importance of ACE2 for viral entry, and the fact that the authors call this into question, the study is very timely and interesting as it may allow scientists to consider further parts of the human body as potential viral entry points, reservoirs or therapeutic structures to name a few. That said, the study has a couple of shortcomings that the authors should address in a revised manuscript.

In the abstract, the authors raise the question if the absence of ACE2 expression in the lung should not imply that other routes of transmission should be explored. While their data seems to be consistent with raising such doubts, it is a 'dangerous' statement as the inability to detect ACE2 in the lung does not mean it is not there or perhaps not required in large copies/surface or perhaps expressed in cells other than the AT2 cells most people suspect. Most infections are detected in the upper airways and the expression data does support this by way of higher ACE2 expression. Incidentally, swabs from the throat and nose often contain very high levels of virus, again underpinning that the primary site of infection are the upper airways. This does not at all mean that the virus does not use ACE2 in e.g. the alveoles and it may indeed not take a lot of infected cells or a lot of ACE2 copies on a cell to set off the lung inflammation and everything that follows from that. Hence, the authors should be more careful in the way they interpret the expression data. Simply because the clinically most severely affected organ is the lung, a more detailed look into cell types of the lung is warranted. The authors have strong ties to pathology groups. It may therefore be possible to obtain autopsy material from Covid19 patients and perform a more detailed investigation into ACE2 expression in these patients. Given that a lot of the patients that show a severe clinical phenotype have strong co-morbidities and many are in fact on ACE inhibitors, looking at autopsies could be even more revealing.

While the protein atlas project is of course a great resource, the authors expand the section of the manuscript dealing with mass spectrometry-based expression analysis in order to confirm or complete the picture of ACE2 expression measured by mRNA or antibodies. So far, the authors only use the human proteome map (Nature 2014) but resources like MaxQB, PAXDB or ProteomicsDB likely by now contain far more expression data.

The section on the cell lines should also be expanded to MS-based proteomic data. There is a lot of data out there on cell line panels and in the aforementioned resources. As a side note, A549 cells are in fact difficult if not impossible to infect with SARS-CoV-2 unless transfected with ACE2. The transfected cell line is highly competent in producing virus, suggesting that indeed ACE2 is important.

Given that the virus requires a host protease for viral entry, the study could be enhanced by analyzing expression of the protease TMPRSS2. That may help to further narrow down the list of potential places where the virus successfully infects cells.

In the discussion, the authors suggest that fecal-oral transmission may be a possible route for infection. How do the authors suggest the virus gets there? Is there any data to suggest that the virus would survive the acidic environment of the stomach? In the hands of this reviewer, it is rather easy to deactivate the virus. Also, would such a route of transmission, while perhaps possible, be

able to explain the relatively rapid transmission rate in all countries, also those with very high levels of hygiene? And if this is so, would the airways not still be at the heart of what ends with severe clinical symptoms?

Much of the discussion is repetition of results. This reviewer suggests that there is indeed more discussion on how the clinical observations can be reconciled with the expression analysis. Some discussion on the drugs that are discussed as potential therapies would be useful in this context too (e.g. ACE inhibitors, protease inhibitors). The authors do end on exactly the right note: "...the need to further explore the route of transmission...to understand the biology of the disease...". Perhaps this is also a good statement for the abstract rather than calling ACE2 into question.

Reviewer #2:

Dear Editor, dear Authors,

the manuscript MSB-20-9610 entitled 'The protein expression profile of ACE2 in human tissues' by Lindskog and colleagues compares different published modalities of ACE2 gene expression and immunostaining in different human tissues. At the core of this analysis, the authors describe that neither reported ACE2 gene expression (bulk or single cell) nor immunostaining of the antibody against ACE2 indicate a clear picture of 'ACE2' in lung tissue or the respiratory tract. In the latter case showing ACE2 being identified in the lung (Hamming et al. 2004), they criticize that the validation quality control did not meet higher standards of the 'International Working Group for Antibody Validation' (IWGAV). Conclusively, the authors directly state and present - in the Human Protein Atlas - that ACE2 protein is not expressed in the lung tissues and hypothesize about an alternative entry of the SARS-CoV-2 virus than through spike protein and ACE2 docking.

Major concerns:

Importantly, the authors raise the question of how no ACE2 protein expression in lung tissue where COVID-19 develops into a fatal disease for humans. However, the different results from all the studies and methods require an improved introduction and analysis according to sensitivity and specificity of these technologies. This aspect is definitely missing and probably also not easy to access or quantify.

Moreover, single cell transcriptomics adds another level of 'specificity' due to single cell types. Again, summing up here over different protocols and methods will miss essential details about sensitivities. Please have a look into recent sc/snRNASeq and benchmarking of the HCA (Ding et al., Mereu et al., NatBiotech 2020).

Secondly, please carefully reassess recent preprints about ACE2 expression in single cells of the lung:

<https://www.biorxiv.org/content/10.1101/742320v1>

<https://arxiv.org/abs/2003.06122>

https://papers.ssrn.com/sol3/papers.cfm?abstract_id=3555145

Although these results are probably all under revisions (not peer-reviewed), the overall general 'statement' is consistent. Specific cell types lowly express ACE2 in the lung. Recently, there are more evidences that an immune response to initial viral SARS-CoV-2 infections might trigger also

ACE2 expression in different cell types.

In my conclusion, we can not exclude alternative non-endosomal (non ACE2) entry routes of this virus in lung cells, but ACE2 exists at low expression in lung cells. I recommend the authors to rethink about their hypothesis after considering the reprints above. Similarly, the editors of MSB should reconsider publication of this manuscript in the context of these preprints.

Reviewer #3:

In this manuscript, the authors use the antibody staining data for ACE2 present in the protein atlas to come to an evaluation of protein expression patterns of the ACE2 receptor in multiple human tissues. Given the relevance of ACE2 as the receptor for the SARS-CoV-2 Spike protein, these data are urgently needed, especially given the contradictory reports present in literature on the subject. Moreover, careful evaluation of the ACE2 expression patterns at the RNA level have recently been published by multiple groups, both at tissue and at single-cell level, but these results also urgently need further validation at the protein level. Hence, the study presented by the authors is timely and could be of great impact for the community and should be published as soon as possible given a careful interpretation and discussion of the data.

Unfortunately, the study presented in its current form fails to offer the careful interpretation and discussion so urgently needed by the community. No one is waiting for an all-or-nothing interpretation of protein expression data such as presented by the authors. In fact, a careful evaluation of protein staining patterns and side-by-side comparison to RNA datasets could lead to a balanced discussion that might really be of huge help to the community, which is really a missed opportunity by the authors.

A more careful discussion should discern more levels of evidence than accepting only the most stringent level of evidence, and dismissing everything that does not meet these criteria as 'no expression detected'. The authors put themselves at a huge risk for false-negative results by this stringent approach and conclusions based on only accepting data that meet the most stringent criteria as 'true' are not going to be very informative. Protein detection methods based on antibodies can suffer from aspecificity, but also from lack of sensitivity - so low protein expression levels might not easily be detected in a way that meets the strict IWGAV criteria - which does not mean that expression is absent. Expression might just be on the low end of the scale, and hard to reproducibly detect. This low level expression might very well still be of great clinical relevance. The authors completely fail to acknowledge and discuss this possibility, severely limiting the relevance and impact of this study as presented at this moment.

The use of the very strict IWGAV recommendations in itself is great - this identifies those tissues and cells where, beyond reasonable doubt, ACE2 protein expression is found. However, the other side of the equation is less clear: those tissues where staining patterns and RNA data do not meet the IWGAV requirements are not necessarily negative - they merely do not meet the very strict criteria! The authors need to consider more levels of evidence, introducing and carefully discussing the category where some evidence points towards expression of ACE2, but the evidence is not conclusive. And this should also be discussed as such - no conclusive evidence for presence OR absence of expression. One such system to present the data with multiple levels-of-evidence could take into account the following categories:

1. Positive staining validated according to gold-standard IWGAV recommendations
2. Positive staining detected with one of the antibodies but not with the other, RNA data support

presence of (low-level) expression

3. No positive staining whatsoever, no RNA expression either.

In the current manuscript, the authors focus fully on category-1 results only. While this is in itself valuable, this yields little novel insights, and does not help the community a whole lot. For instance, the authors currently conclude regarding expression in lung that there is 'no detectable expression in lung or respiratory epithelia.' (first paragraph Discussion section). This then leads the authors to speculate about alternative receptors for SARS-CoV-2 in respiratory epithelia. The correct conclusion, however, should be that 'we cannot offer conclusive data for absence or presence of ACE2 expression in lung and respiratory epithelia, and more studies are urgently needed.' Even if this provides no yes-or-no answer, this is what the community needs to hear, and what the authors should present on the basis of their data.

In addition to this overall criticism, I have several additional remarks and suggestions for this manuscript, hoping to improve its quality. The authors should be given the opportunity to rephrase their data presentation and discussion, as this study is - when offered in a balanced way - very urgently needed by the community!

My additional comments are:

Page 3 - Northern blotting is a quite low-sensitive method. The methods mentioned here in the beginning of the second paragraph will only detect high expression, and are less suited for detection of low levels of ACE2 mRNA expression. So this does not inform on the distinction between category ii and iii as mentioned above. Please rephrase for more careful interpretation of these literature data - and move to introduction rather than results section?

Page 3 - it would make sense to mention the Human Cell Atlas consortium along with the other international consortia, as the data generated by the HCA are at the single-cell resolution and very informative with regards to the goals of this study. This also still seems introduction.

Page 4 - last sentence first paragraph - please mention that the expression analysed is of ACE2. The presentation of the protein data needs reformatting - please see comments to figure 1 below. Absence of expression is hard to conclude, again - please discuss more carefully, especially if the underlying data are not being presented to the audience (a single cropped picture for one Ab staining lacking any and all detail does not meet any criterion at all)

Page 4 - second paragraph - it is unclear why only a single study from literature is discussed, while other studies have already claimed absence of ACE2 expression in respiratory epithelia - this should be discussed as well. This is also not discussing the data presented by the authors but a 10-year old study from literature - this seems better placed in the discussion section? What is the point of extensively discussing why other people's data are of less quality when the authors fail to accurately present and discuss their own staining data (a heatmap is aggregate statistics, not staining data). See also next comments

Page 5 -> after careful discussion on the substandard quality of a single 10-year old paper the authors finally start to present their own staining data. Unfortunately, a single snapshot is shown (figure 2) per location/tissue/cell-type, and only stained by 1 Ab. No data are available on age, health status, smoking status and sex of the tissue donor, metadata that are highly relevant to ACE2 expression levels. So it is impossible to know whether these results can be generalized, and the reader is just stuck with a highly uninformative selection of tissue stainings. For careful evaluation of the claims made by the authors, multiple tissue sections stained with multiple antibodies (per location/organ/cell-type) need to be shown, maybe in supplement but ready for inspection by the reader. Also the correlation between the 2 stainings needs to be shown.

Page 5 - mass spec -> this is also a low-sensitivity approach, so low expression of ACE2 will not be detected. This needs to be discussed: it adds only to evidence for high expression, but cannot

distinguish between absence of expression versus low levels of expression

Page 6 - scRNA-seq data -> The data presented here are of interest, but the fashion in which they are presented is extremely poor. As the authors claim, ACE2 expression in lung is low, and showing a UMAP plot with individually stained cells does not allow any inspection of the data in a meaningful way. The authors are well-aware of how to generate heatmaps (figure 1) - why not do the same for the scRNA-Seq data from lung - that would allow for a great comparison. These datasets have all recently been published, or re-analyzed for ACE2 expression, and these papers, as referenced by the authors, do offer a far better presentation of the data in a fashion that actually allows meaningful interpretation and comparison to the protein expression data, which would be very relevant to the community. Clearly, the ACE2 receptor has low expression in lung, but in certain cell types RNA expression is observed in 5-10% of the cells, which clearly warrants comparison to the protein expression data for meaningful interpretation. The way the authors choose to show these data however, is not helpful at all, and does not allow any comparison to protein data. In addition, these papers also contain RNA expression data of the cells of other tissues, including those where the authors do observe positive protein expression. Also here, RNA to protein comparison would be extremely valuable. This part of the manuscript really needs to be redone properly, with the intent to compare RNA and protein results across tissues in a meaningful way. The scRNA-Seq data are freely available, so there's no reason why the authors should not take more care in the presentation of their results here.

Page 6/7 - cell lines - please remove this part, it really does not add anything compared to the discussion of human tissues and cells analysed by IHC or (sc)RNA-Seq.

Discussion - as mentioned before the claim of the authors that there is no expression of ACE2 in lung is outrageous. There is no evidence for ACE2 expression when applying the strict IWGAV recommendations. However, that does NOT mean that there is no expression whatsoever. The authors fail to provide evidence for this claim, and this therefore needs to be reconsidered. The authors urgently need to discuss the levels of evidence for ACE2 expression, which is high for certain tissues, low for some others including lung, and absent in some others. Also, some tissues are completely ignored by the authors, including epithelial of the superficial conjunctiva in eye and respiratory epithelia in nose - which at RNA level have high expression levels. A discussion of the limitation of antibody-based protein detection would also seem in place, which might well be combined with a levels-of-evidence discussion.

Figure 1 - showing this heatmap without the underlying data (quantification, variability of measurement, within-study variation, how many sections/views analysed, etc) is not very meaningful. Also, this literature overview seems incomplete, with unclear criteria for inclusion or exclusion of individual studies and datasets. Why are single-cell datasets not included in this analysis? Why are Northern blot and qRT-PCR datasets included when we do have RNA-Seq datasets available? How are RNA and protein data compared, what does 'high' on protein data really mean, and how well can that be compared to 'high' on RNA data? Why was the cut-off set at 3-10% 11-49% and 50-100%? What happens if quartiles are shown? Is the distribution of observations similar for RNA and protein data? Why is up to 2% of the maximum value considered negative? If this compares to single-cell data, this would mean that 2% positive cells would be considered negative - what is the rationale?

Figure 2 - please show more replicates per tissue, showing both antibodies used and the correlation between their staining patterns as a function of cell type and tissue studied. A more exact localization for some tissues would be great. Lung -> where in lung? Nasopharynx -> where exactly (to be able to compare to the scRNA-Seq data)

Figure 3 - please redo this figure in a similar fashion to figure 1 to allow a more accurate interpretation, and do include nasal data.

Some references need to be updated, and some more relevant studies have been published during the review process, but that is of course very understandable. Please update the references to

include the latest data in the field.

All in all this is a study with great potential, and one that is urgently needed by the field, but it needs a more balanced presentation and discussion of the data.

Answers to Editorial comments

Reviewer #1 raises a very good point, which refers to the potentially important insights that could be derived by analyzing data from covid-19 patient samples. In case you have access to such material we would strongly encourage you to perform these analyses, as they would significantly enhance the conclusiveness and impact of the study and could address questions including the effect of immune responses on ACE2 levels, the link to the expression of the protease TMPRSS2 etc.

We unfortunately do not have access to samples from covid-19 patients nor could obtain it within a reasonable time-frame, and therefore feel that analysis of such material would be better suited as a follow-up study. Here, we aim to provide a comprehensive overview of ACE2 expression under normal circumstances based on different methodologies. In order to expand the present investigation, we have however added additional normal tissue samples including eye, larger sections of upper airway epithelia from >8 different individuals, and a well-characterized cohort of 360 normal lung samples.

Reviewer #1 mentions that further proteomics resources need to be considered in the analyses.

While Reviewer #1 is correct that we could do an extended analysis on more proteomics datasets, we also agree with the comment from Reviewer #3 that mass spectrometry represents a low-sensitivity approach not well suited to search for a protein that in some tissues may be low abundant. However, in order to provide a comprehensive overview of ACE2 expression taking into consideration different approaches, we have decided to look into more proteomics datasets using online databases (PaxDB and ProteomicsDB) and provided a new summary of the results. Additionally, the antibodies used for immunohistochemistry have also been validated by Western blot, presented in a new figure. The discussion has been expanded, bringing up disadvantages and advantages of using different methods for the analysis of protein expression, including antibody-based proteomics and mass spectrometry.

A more balanced interpretation and discussion of alternative explanations and hypotheses is warranted to provide a more informative view on what the observed low ACE2 protein levels might mean for covid-19 infection.

We agree and have now expanded the discussion on this subject.

As reviewers #2 and #3 recommend, a more thorough comparison and discussion of RNA-seq and scRNA-seq data needs to be included.

This has been considered in the revised version of the manuscript. The results from RNA expression data based on three different datasets (HPA, GTEx, and FANTOM5) have now been presented and discussed more in detail. We have also added additional analyses of scRNA-seq data, including data from upper airways, ileum, kidney and testis. Finally, we have also added a

more detailed comparison and discussion of the different methods, both in the results section and in the discussion.

Answers to Reviewer #1

In the present manuscript, Hikmet et al analyse the (protein) expression of ACE2 using antibodies in 78 different human cell types. The study is motivate by the current SARS-CoV-2 pandemic and the presumed essential role of ACE2 in the infection process. Given the perceived importance of ACE2 for viral entry, and the fact that the authors call this into question, the study is very timely and interesting as it may allow scientist to consider further parts of the human body as potential viral entry points, reservoirs or therapeutic structures to name a few. That said, the study has couple of shortcomings that the authors should address in a revised manuscript.

We thank the reviewer for acknowledging the importance of the study and the insightful comments, which we have addressed in a revised version of the manuscript.

In the abstract, the authors raise the question if the absence of ACE2 expression in the lung should not imply that other routes of transmission should be explored. While their data seems to be consistent with raising such doubts, it is a 'dangerous' statement as the inability to detect ACE2 in the lung does not mean it is not there or perhaps not required in large copies/surface or perhaps expressed in cells other than the AT2 cells most people suspect. Most infections are detected in the upper airways and the expression data does support this by way of higher ACE2 expression. Incidentally, swaps from the throat and nose often contain very high levels of virus, again underpinning that the primary side of infection are the upper airways. This does not at all mean that the virus does not use ACE2 in e. g. the alveoles and it may indeed not take a lot of infected cells or a lot of ACE2 copies on a cell to set off the lung inflammation and everything that follows from that. Hence, the authors should be more careful in the way they interpret the expression data.

We agree that it should be discussed further in the manuscript that ACE2 could be expressed in other cell types than AT2, e.g. upper airway epithelia, or in lower amounts. Since the previous analysis only contained samples from 3 individuals for each tissue in a tissue microarray format, we have now expanded the analysis to also include more samples from bronchus and nasopharynx, using larger sections. These new results have been added to the revised version of the manuscript. Furthermore, we have also looked more into scRNA-seq data from upper airways and added a more detailed discussion on the ACE2 expression in these cell types. Still, we do feel that alternative routes of transmission needs to be discussed, which we now have addressed more in detail.

Simply because the clinically most severely affected organ is the lung, a more detailed look into cell types of the lung is warranted. The authors have strong ties to pathology groups. It may therefore be possible to obtain autopsy material form Covid19 patients and perform a more detailed investigation into ACE2 expression in these patients. Given that a lot of the patients that show a severe clinical phenotype have strong co-

morbidities and many are in fact on ACE inhibitors, looking at autopsies could be even more revealing.

Unfortunately, we currently have no access to such material. In fact, only a few autopsies of Covid-19 patients have been performed at Uppsala Akademiska Hospital, the pathology department connected to our institute. We agree that this would be a very interesting follow-up study, and we are looking into the possibility to gain access to such samples for future projects. In the present however, the objective is to present a comprehensive overview of ACE2 expression in normal human tissues. In order to expand the current analysis, we have however performed additional experiments on >300 normal lung samples. This novel data is presented in the revised version of the manuscript, together with a more detailed discussion of the results.

While the protein atlas project is of course a great resource, the authors expand the section of the manuscript dealing with mass spectrometry-based expression analysis in order to confirm or complete the picture of ACE2 expression measured by mRNA or antibodies. So far, the authors only use the human proteome map (Nature 2014) but resources are MaxQB, PAXDB or ProteomicsDB likely by now contain far more expression data.

While it is important to study the expression of ACE2 using different methods, mass spectrometry-based analysis represents a low-sensitivity approach not well suited to search for a protein that in some tissues may be low abundant. This is also pointed out by another reviewer. However, as we do think it is important to validate the ACE2 expression using different approaches both on the mRNA and protein level, we decided to look into more proteomics datasets and in the revised version of the manuscript, we provide a new summary of the mass spectrometry-based results from PaxDB and proteomicsDB according your suggestions. Although these resources gave more information on ACE2 expression among tissues, the data came from untargeted MS studies and it cannot be ruled that low expression were not detected. Additionally, the antibodies used for immunohistochemistry have also been validated by Western blot, presented in a new figure. The discussion has been expanded, bringing up disadvantages and advantages of using different methods for the analysis of protein expression, including antibody-based proteomics and mass spectrometry.

The section on the cell lines should also be expanded to MS-based proteomic data. There is a lot of data out there on cell line panels and in the aforementioned resources. As a side note, A549 cells are in fact difficult if not impossible to infect with SARS-CoV-2 unless transfected with ACE2. The transfected cell line is highly competent in producing virus, suggesting that indeed ACE2 is important.

We agree that the analysis on cell lines is limited, but instead of expanding it, we have decided to remove it in the revised version of the manuscript, as suggested by another reviewer. Analysis of cell lines is out of the scope of the present investigation, as the main objective is to provide a comprehensive overview of ACE2 expression across different normal tissues and organs.

Given that the virus requires a host protease for viral entry, the study could be enhanced by analyzing expression of the protease TMPRSS2. That may help to further narrow down the list of potential places where the virus successfully infects cells.

Analysis of TMPRSS2 expression would indeed lead to additional insights on potential sites for virus entry. It should however be noted that a recent study could only show co-expression of TMPRSS2 and ACE2 in a subset of cells (Sugnak et al, Nature Medicine, 2020), suggesting that the virus uses alternative pathways such as CTSB or CTSL, or possibly other proteases. In order to investigate this in a thorough way, we feel that it is better suited as a follow-up study, where we can look into the co-expression of ACE with TMPRSS2, CTSB and CTSL in various human cell types using different methods for co-localization.

In the discussion, the authors suggest that fecal-oral transmission may be a possible route for infection. How do the authors suggest the virus gets there? Is there any data to suggest that the virus would survive the acidic environment of the stomach? In the hands of this reviewer, it is rather easy to deactivate the virus. Also, would such a route of transmission, while perhaps possible, be able to explain the relatively rapid transmission rate in all countries, also those with very high levels of hygiene? And if this is so, would the airways not still be at the heart of what ends with severe clinical symptoms?

This is an important point that should be discussed further. While it is clear that Covid-19 leads to severe respiratory symptoms, the exact pathophysiological mechanisms of human transmission and infection are still unclear. It is also not known how high expression of viral receptors that would be necessary for infection.

Recent clinical descriptions point at an inflammatory reaction both systemically and in the lungs, that in severe cases may lead to acute respiratory disease syndrome (ARDS) and death. Still, the overall clinical picture is not consistent with typical ARDS and it remains to be elucidated if the symptoms are driven by local virus infection of the cells in the lung, or caused by secondary effects. Patients with Covid-19 seem to have a larger inflammatory response compared to other viral respiratory diseases, and effects on coagulation and vascular dysfunction with high risk of thrombosis have also been observed. This would fit with the theory that the SARS-CoV-2 virus infects the upper airways and then spreads to the rest of the body, suggesting that the damage to the lungs could be a secondary effect.

Fecal-oral transmission does seem possible, which is supported by a recent study showing that the virus can productively infect human gut enterocytes (Lamers et al, Science 2020). Other alternatives to ACE2-mediated infection for SARS-CoV-2 host cell entry are the use of another yet to be confirmed receptor in the lungs, or non-receptor dependent infection. Large studies using samples from Covid-19 patients are needed in order to confirm the histological co-localization of the virus with ACE2 and other SARS-CoV-2 related proteins, and correlate the

findings with the disease progression. An extended discussion on this has been added to the revised version of the manuscript.

Much of the discussion is repetition of results. This reviewer suggests that there is indeed more discussion on how the clinical observations can be reconciled with the expression analysis. Some discussion on the drugs that are discussed as potential therapies would be useful in this context too (e.g. ACE inhibitors, protease inhibitors). The authors do end on exactly the right note: "...the need to further explore the route of transmission...to understand the biology of the disease...". Perhaps this is also a good statement for the abstract rather than calling ACE2 into question.

In line with the previous comment, we agree that an extended discussion on how the clinical manifestation of the disease could relate to expression levels of ACE2 in different human tissues is of relevance for the reader. This has now been added to the revised version of the manuscript. With regard to potential drugs however, there has been new research in this area during the last month, including three recent papers showing no evidence for a detrimental effect of Covid-19 with the use of ACE2 or ARB inhibitors (Mehra et al, NEJM, 2020; Reynolds et al, NEJM, 2020; Mancina et al, NEJM, 2020). We therefore feel that it is out of the scope of the present investigation to add an extended discussion on potential therapies.

Answers to Reviewer #2

The manuscript MSB-20-9610 entitled 'The protein expression profile of ACE2 in human tissues' by Lindskog and colleagues compares different published modalities of ACE2 gene expression and immunostaining in different human tissues. At the core of this analysis, the authors describe that neither reported ACE2 gene expression (bulk or single cell) nor immunostaining of the antibody against ACE2 indicate a clear picture of 'ACE2' in lung tissue or the respiratory tract. In the latter case showing ACE2 being identified in the lung (Hamming et al. 2004), they criticize that the validation quality control did not meet higher standards of the 'International Working Group for Antibody Validation' (IWGAV). Conclusively, the authors directly state and present - in the Human Protein Atlas - that ACE2 protein is not expressed in the lung tissues and hypothesize about an alternative entry of the SARS-CoV-2 virus than through spike protein and ACE2 docking.

We thank the reviewer for summarizing the key points of the study.

Major concerns:

Importantly, the authors raise the question of how no ACE2 protein expression in lung tissue where COVID-19 develops into a fatal disease for humans. However, the different results from all the studies and methods require an improved introduction and analysis according to sensitivity and specificity of these technologies. This aspect is definitely missing and probably also not easy to access or quantify.

We agree that the different studies and methods used could be summarized more clearly. In the revised version of the manuscript, we have added more details to the results, and have expanded the discussion comparing the results observed by different studies, including a description of different advantages and disadvantages with the various technological approaches.

Moreover, single cell transcriptomics adds another level of 'specificity' due to single cell types. Again, summing up here over different protocols and methods will miss essential details about sensitivities. Please have a look into recent sc/snRNASeq and benchmarking of the HCA (Ding et al., Mereu et al., NatBiotech 2020).

We thank the reviewer for the suggestion. In the revised version of the manuscript, we have added a section on recent HCA efforts, including standardization and benchmarking.

In order to re-analyze scRNA-seq datasets in line with standard procedures, we have followed the Seurat tutorial available on the satijalab.org/seurat website, especially the major components of clustering workflow. Furthermore, we have taken into consideration the recommendations of Luecken and Theis's review "Current best practices in single-cell RNA-seq : a tutorial." (Luecken et al, MSB, 2019).

Secondly, please carefully reassess recent preprints about ACE2 expression in single cells of the lung:

<https://www.biorxiv.org/content/10.1101/742320v1>

<https://arxiv.org/abs/2003.06122>

https://papers.ssrn.com/sol3/papers.cfm?abstract_id=3555145

The dataset for the first preprint is not yet available, however, the second preprint by Sungnak et al. was referenced and briefly mentioned also in our first submission. In the revised version of the manuscript, we have now included the same datasets that were used in the Sungnak paper in our re-analysis of scRNA-seq data from upper airways. The results are presented and compared with an extended analysis using immunohistochemistry on tissue samples from nasopharynx and bronchus.

Although these results are probably all under revisions (not peer-reviewed), the overall general 'statement' is consistent. Specific cell types lowly express ACE2 in the lung. Recently, there are more evidences that an immune response to initial viral SARS-CoV-2 infections might trigger also ACE2 expression in different cell types.

While many recent studies using scRNA-seq, both published and non peer-reviewed, point at a very low expression of ACE2 in AT2 cells, there are however also differences between the datasets, as seen in the studies re-analyzed in the present investigation. The Human Cell Landscape (Han et al, Nature, 2020) shows highest expression (0.3%) in a mixed cluster representing AT1/AT2 cells, together with very low levels (<0.1%) in other clusters representing AT1 or AT2 cells. No expression was observed in secretory or ciliated cells, that would fit with the theory that higher expression of ACE2 would be observed in upper airway epithelia. The other two analyzed lung datasets (Reyfman et al, Am J Respir Crit Care Med, 2019; Viera Braga et al, Nat Med, 2019) also point at very low expression in AT2 cells, but the expression differed in the other lung cell types. In Reyfman et al the expression was almost equally high in AT1 cells, secretory cells and granulocytes, while Viera Braga showed no expression in these cell types but instead expression in ciliated cells. In summary, we agree that a more detailed discussion on different scRNA-seq datasets and the expression in lung is needed, also bringing up recent studies on immune response triggered ACE2 expression. This has been added to the revised version of the manuscript.

It is also important to note that although several studies based on scRNA-seq suggest certain cell types that seem to express ACE2 at lower levels, it is not certain how much from this that can be inferred in terms of physiological and functional phenotypes on the protein level. In the revised version of the manuscript, we have expanded the discussion to also highlight the importance of studying protein levels, as well as advantages and disadvantages of different technologies used for analysis of cell type expression, and possible reasons for discordance between various datasets.

In my conclusion, we can not exclude alternative non-endosomal (non ACE2) entry routes of this virus in lung cells, but ACE2 exists at low expression in lung cells. I

recommend the authors to rethink about their hypothesis after considering the reprints above. Similarly, the editors of MSB should reconsider publication of this manuscript in the context of these preprints.

We agree that it is of relevance for the reader with an extended discussion on possible routes of host cell entry based on ACE2 expression in different cell types according to both the present and other recent studies. It is also important to put this in context with clinical observations of Covid-19 patients, and how the expression of ACE2 in different cells and organs fits with various hypotheses of host cell entry. This has now been added to the revised version of the manuscript.

Answers to Reviewer #3

In this manuscript, the authors use the antibody staining data for ACE2 present in the proteinaatlas to come to an evaluation of protein expression patterns of the ACE2 receptor in multiple human tissues. Given the relevance of ACE2 as the receptor for the SARS-CoV-2 Spike protein, these data are urgently needed, especially given the contradictory reports present in literature on the subject. Moreover, careful evaluation of the ACE2 expression patterns at the RNA level have recently been published by multiple groups, both at tissue and at single-cell level, but these results also urgently need further validation at the protein level. Hence, the study presented by the authors is timely and could be of great impact for the community and should be published as soon as possible given a careful interpretation and discussion of the data.

We thank the reviewer for acknowledging the value of the study.

Unfortunately, the study presented in its current form fails to offer the careful interpretation and discussion so urgently needed by the community. No one is waiting for an all-or-nothing interpretation of protein expression data such as presented by the authors. In fact, a careful evaluation of protein staining patterns and side-by-side comparison to RNA datasets could lead to a balanced discussion that might really be of hige help to the community, which is really a missed opportunity by the authors.

We agree that a more detailed presentation of the results from both the present investigation and other recent studies, together with an expanded discussion would be helpful for the reader. In the revised version of the manuscript, we have added novel immunohistochemistry data from >300 normal lung samples together with more samples from nasopharynx, bronchus and eye. We have also analyzed both antibodies with Western blot, expanded the section on mass spectrometry, expanded the section on scRNA-seq, including re-analysis of more datasets, as well as changed the presentation of the results for more clarity. We have also added an extended discussion on comparison between the results observed by various datasets, including advantages and disadvantages of different technologies for studying cell or tissue type expression.

A more careful discussion should discern more levels of evidence than accepting only the most stringent level of evidence, and dismissing everything that does not meet these criteria as 'no expression detected'. The authors put themselves at a huge risk for false-negative results by this stringent approach and conclusions based on only accepting data that meet the most stringent criteria as 'true' are not going to be very informative. Protein detection methods based on antibodies can suffer from aspecificity, but also from lack of sensitivity - so low protein expression levels might not easily be detected in a way that meets the strict IWGAV criteria - which does not mean that expression is absent. Expression might just be on the low and of the scale, and hard to reproducibly detect. This low level expression might very well still be of great clinical relevance. The

authors completely fail to acknowledge and discuss this possibility, severely limiting the relevance and impact of this study as presented at this moment.

This is an important point and we agree that a more detailed discussion around the use of antibodies for detection of low-abundant proteins is necessary. In the revised version of the manuscript, a more careful presentation of the results is provided, including findings in tissues that are suggested to express lower amounts of ACE2. We have also added a more detailed discussion around findings observed by various technologies used for analysis of cell or tissue type expression.

The use of the very strict IWGAV recommendations in itself is great - this identifies those tissues and cells where, beyond reasonable doubt, ACE2 protein expression is found. However, the other side of the equation is less clear: those tissues where staining patterns and RNA data do not meet the IWGAV requirements are not necessarily negative - they merely do not meet the very strict criteria! The authors need to consider more levels of evidence, introducing and carefully discussing the category where some evidence points towards expression of ACE2, but the evidence is not conclusive. And this should also be discussed as such - no conclusive evidence for presence OR absence of expression. One such system to present the data with multiple levels-of-evidence could take into account the following categories:

- 1. Positive staining validated according to gold-standard IWGAV recommendations**
- 2. Positive staining detected with one on the antibodies but not with the other, RNA data support presence of (low-level) expression**
- 3. No positive staining whatsoever, no RNA expression either.**

This is a good and helpful suggestion, and we have taken this into consideration in the revised version of the manuscript, providing a more careful and detailed presentation of how protein expression observed by IHC by one, both or none of the antibodies on a cell type specific level link with RNA expression levels, including expression levels that are low or close to cutoff. Both the figures and the text have been updated, and new supplementary data has been added.

In the current manuscript, the authors focus fully on category-1 results only. While this is in itself valuable, this yields little novel insights, and does not help the community a whole lot. For instance, the authors currently conclude regarding expression in lung that there is 'no detectable expression in lung or respiratory epithelia.' (first paragraph Discussion section). This then leads the authors to speculate about alternative receptors for SARS-CoV-2 in respiratory epithelia. The correct conclusion, however, should be that 'we cannot offer conclusive data for absence or presence of ACE2 expression in lung and respiratory epithelia, and more studies are urgently needed.' Even if this provides no yes-or-no answer, this is what the community needs to hear, and what the authors should present on the basis of their data.

In the originally submitted version of the manuscript, we focused only on results where we confidently could confirm distinct expression of ACE2 beyond reasonable doubt, consistent with

RNA expression levels and results using two different antibodies. We however agree that there are tissues and cell types where the conclusion is less clear, and that it is important for the reader with a more detailed discussion also around these findings. This has been added to the revised version of the manuscript.

In addition to this overall criticism, I have several additional remarks and suggestions for this manuscript, hoping to improve its quality. The authors should be given the opportunity to rephrase their data presentation and discussion, as this study is - when offered in a balanced way - very urgently needed by the community!

My additional comments are:

Page 3 - Northern blotting is a quite low-sensitive method. The methods mentioned here in the beginning of the second paragraph will only detect high expression, and are less suited for detection of low levels of ACE2 mRNA expression. So this does not inform on the distinction between category ii and iii as mentioned above. Please rephrase for more careful interpretation of these literature data - and move to introduction rather than results section?

In the revised version of the manuscript, we have removed the Northern blot data from results and instead mention it in the introduction.

Page 3 - it would make sense to mention the Human Cell Atlas consortium along with the other international consortia, as the data generated by the HCA are at the single-cell resolution and very informative with regards to the goals of this study. This also still seems introduction.

We thank the reviewer for the suggestion. In the revised version of the manuscript, we have added information on recent HCA efforts, including standardization and benchmarking.

Page 4 - last sentence first paragraph - please mention that the expression analysed is of ACE2. The presentation of the protein data needs reformatting - please see comments to figure 1 below. Absence of expression is hard to conclude, again - please discuss more careful, especially if the underlying data are not being presented to the audience (a single cropped picture for one Ab staining lacking any and all detail does not meet any criterion at all).

In the revised version of the manuscript, we have added more detailed figures on both the RNA and protein expression analysis of ACE2, together with more careful presentation and discussion of the results. Please also see the response to the comments to Figure 1 below.

Page 4 - second paragraph - it is unclear why only a single study from literature is discussed, while other studies have already claimed absence of ACE2 expression in respiratory epithelia - this should be discussed as well. This is also not discussing the

data presented by the authors but a 10-year old study from literature - this seems better placed in the discussion section? What is the point of extensively discussing why other people's data are of less quality when the authors fail to accurately present and discuss their own staining data (a heatmap is aggregate statistics, not staining data). See also next comments

The reason for extensively discussing the results by Hamming et al from 2004 is the fact that this older study is referred to in many studies published recently, often together with conclusions stating that it is proven that ACE2 is highly expressed in AT2 cells. As we believe that many of the findings from the study by Hamming et al are based on unspecific antibody binding, we think that is of high relevance for the scientific community to discuss these previous findings in detail. We however agree that the discussion should be balanced with more details around both our own results and other recent studies. This has been considered in the revised version of the manuscript. The previous Figure 1 has been removed and replaced with a more detailed presentation of RNA expression datasets from HPA, GTEx and FANTOM5, showing the exact values instead of using specific cut-offs. The results from other datasets such as Northern blot, QRT-PCR and IHC data from Hamming et al are now only described in the text, and the IHC data from the present investigation have both been expanded and are discussed more in detail. Please also see the response to the comments to Figure 2 below.

Page 5 -> after careful discussion on the substandard quality of a single 10-year old paper the authors finally start to present their own staining data. Unfortunately, a single snapshot is shown (figure 2) per location/tissue/cell-type, and only stained by 1 Ab. No data are available on age, health status, smoking status and sex of the tissue donor, metadata that are highly relevant to ACE2 expression levels. So it is impossible to know whether these results can be generalized, and the reader is just stuck with a highly uninformative selection of tissue stainings. For careful evaluation of the claims made by the authors, multiple tissue sections stained with multiple antibodies (per location/organ/cell-type) need to be shown, maybe in supplement but ready for inspection by the reader. Also the correlation between the 2 stainings needs to be shown.

The reason for only showing representative images from one antibody, without adding details on age, gender etc. for each of the individual samples is because the selected images shown in the original version of the manuscript correspond to high-resolution images and data that is publicly available on www.proteinatlas.org, where the reader can carefully examine all details on the results. We however agree that this is not clearly stated in the manuscript, and that it would be more helpful for the scientific community with a thorough presentation of the results. In the revised version of the manuscript, we have analyzed more samples for each tissue type, stained with the two different antibodies on consecutive sections. We have added more detailed IHC images from both antibodies, and also provide a supplementary table listing the exact number of samples analyzed for each tissue type, together with available patient data. In the extended cohort of >300 human lung samples, data on performance status, smoking and other clinical parameters are provided, however, for the other analyzed tissues, we only have information on

age and gender according since these tissue samples are derived from anonymized biobank material.

Page 5 - mass spec -> this is also a low-sensitivity approach, so low expression of ACE2 will not be detected. This needs to be discussed: it adds only to evidence for high expression, but cannot distinguish between absence of expression versus low levels of expression.

We agree that mass spectrometry is a low-sensitivity approach, but still think it adds important value to compare the findings on expression levels using different methods. As suggested by another reviewer, we have in the revised version of the manuscript looked into MS databases and provided a more thorough description and discussion of the results, including a discussion around different advantages and disadvantages using different technologies for analysis of cell and tissue level expression.

Page 6 - scRNA-seq data -> The data presented here are of interest, but the fashion in which they are presented is extremely poor. As the authors claim, ACE2 expression in lung is low, and showing a UMAP plot with individually stained cells does not allow any inspection of the data in a meaningful way. The authors are well-aware of how to generate heatmaps (figure 1) - why not do the same for the scRNA-Seq data from lung - that would allow for a great comparison. These datasets have all recently been published, or re-analyzed for ACE2 expression, and these papers, as referenced by the authors, do offer a far better presentation of the data in a fashion that actually allows meaningful interpretation and comparison to the protein expression data, which would be very relevant to the community. Clearly, the ACE2 receptor has low expression in lung, but in certain cell types RNA expression is observed in 5-10% of the cells, which clearly warrants comparison to the protein expression data for meaningful interpretation. The way the authors choose to show these data however, is not helpful at all, and does not allow any comparison to protein data. In addition, these papers also contain RNA expression data of the cells of other tissues, including those where the authors do observe positive protein expression. Also here, RNA to protein comparison would be extremely valuable. This part of the manuscript really needs to be redone properly, with the intent to compare RNA and protein results across tissues in a meaningful way. The scRNA-Seq data are freely available, so there's no reason why the authors should not take more care in the presentation of their results here.

In the revised version of the manuscript, we have replaced the previous figure in order to present the scRNA-seq data more clearly for the reader. This includes re-analysis of datasets from airway epithelia that were recently published, and addition of other relevant tissues such as kidney, testis and ileum, in order to compare the observed expression levels in the respiratory system with other organs. We have also more carefully addressed how ACE2 levels in different cell types based on scRNA-seq relate to body-wide analysis of expression levels using other methods, such as IHC.

Page 6/7 - cell lines - please remove this part, it really does not add anything compared to the discussion of human tissues and cells analysed by IHC or (sc)RNA-Seq.

We agree that analysis of cell lines is out of the scope of the present investigation, as the main objective is to provide a comprehensive overview of ACE2 expression in different normal tissues. The cell line section has been removed in the revised version of the manuscript.

Discussion - as mentioned before the claim of the authors that there is no expression of ACE2 in lung is outrageous. There is no evidence for ACE2 expression when applying the strict IWGAV recommendations. However, that does NOT mean that there is no expression whatsoever. The authors fail to provide evidence for this claim, and this therefore needs to be reconsidered. The authors urgently need to discuss the levels of evidence for ACE2 expression, which is high for certain tissues, low for some others including lung, and absent in some others. Also, some tissues are completely ignored by the authors, including epithelial of the superficial conjunctiva in eye and respiratory epithelia in nose - which at RNA level have high expression levels. A discussion of the limitation of antibody-based protein detection would also seem in place, which might well be combined with a levels-of-evidence discussion.

As discussed in previous comments above, we agree with the reviewer that a more thorough discussion on the results in tissues that are suggested to have low expression of ACE2 is needed. We also agree with the fact that expanding the analysis with more relevant tissues would give a more complete overview of the ACE2 expression across the human body. In the revised version of the manuscript, we have therefore added novel results on protein expression in eye, more samples and regions of respiratory epithelia in addition to the tissue microarray samples from these tissues analyzed previously, and also performed an in-depth analysis of human lung using a well-characterized cohort of 360 normal lung samples.

Figure 1 - showing this heatmap without the underlying data (quantification, variability of measurement, within-study variation, how many sections/views analysed, etc) is not very meaningful. Also, this literature overview seems incomplete, with unclear criteria for inclusion or exclusion of individual studies and datasets. Why are single-cell datasets not included in this analysis? Why are Northern blot and qRT-PCR datasets included when we do have RNA-Seq datasets available? How are RNA and protein data compared, what does 'high' on protein data really mean, and how well can that be compared to 'high' on RNA data? Why was the cut-off set at 3-10% 11-49% and 50-100%? What happens if quartiles are shown? Is the distribution of observations similar for RNA and protein data? Why is up to 2% of the maximum value considered negative? If this compares to single-cell data, this would mean that 2% positive cells would be considered negative - what is the rationale?

As discussed above, we have now removed this heatmap and replaced it with a new figure with bar plots detailing the exact values of RNA expression levels based on HPA, GTEx and FANTOM5.

Figure 2 - please show more replicates per tissue, showing both antibodies used and the correlation between their staining patterns as a function of cell type and tissue studied. A more exact localization for some tissues would be great. Lung -> where in lung? Nasopharynx -> where exactly (to be able to compare to the scRNA-Seq data).

We have now added more detailed images from both antibodies in the main figure, as well as in a supplementary figure. In the same figures, we present RNA expression levels for the tissues shown, based on normalized expression of HPA, GTEx and FANTOM5 data. Unfortunately, it is not possible to obtain information on the exact localization of sampling, as the analysis is based on anonymous biobank material with limited information available. All tissues are however thoroughly examined by a pathologist, and based on histology, it is possible to distinguish different structures unique to certain tissues or cell types. This has been described more carefully when presenting the results. We have also added a detailed supplementary table providing protein expression levels across all analyzed >150 different cell types using the two antibodies.

Figure 3 - please redo this figure in a similar fashion to figure 1 to allow a more accurate interpretation, and do include nasal data.

We thank the reviewer for this suggestion, and have updated the scRNA-seq figure accordingly in the revised version of the manuscript.

Some references need to be updated, and some more relevant studies have been published during the review process, but that is of course very understandable. Please update the references to include the latest data in the field.

The reference list has now been updated, taking into consideration both preprints that have been published during the review process, and the addition of other relevant recent preprints.

All in all this is a study with great potential, and one that is urgently needed by the field, but it needs a more balanced presentation and discussion of the data.

Thank you for sending us your revised manuscript. We have now heard back from the two referees who were asked to review the revised study. As you will see below, both referees are satisfied with the modifications made and are supportive of publication. Reviewer #3 only raises two minor points, which we would ask you to address in a minor revision

We would also ask you to address the following remaining editorial points:

REFEREE REPORTS

Reviewer #1:

The authors have adequately addressed my concerns. Not everything can be resolved and, sharing the results at this time would seem more important than working out further details. I therefore support publication of the work.

Reviewer #3:

First of all I would like to apologize for the late review - the volume of review requests has increased substantially and unfortunately I was unable to review the resubmission by the indicated time. This is especially unfortunate given the excellent job the authors have done on the revision of this manuscript. It is now a very high-quality manuscript that needs very little if any additional editing. The authors accurately addressed my main concerns and most of the minor concerns with the initial submission, and it's great to see that the manuscript is now very convincing, well balanced and thorough. In addition, I would like to mention the attractive visualization, which really helps getting the message across.

There's 2 small, remaining points that the authors could address:

- the staining pattern in most barrier tissues is very similar, with a highly apical staining within the epithelium. In all cases, the HPA000288 Ab seems to give a stronger staining of this particular staining pattern than the MAB933. In the respiratory epithelia, this same staining pattern is observed only with the HPA000288 Ab, and is observed in all subjects in the nasal epithelium (with exception of the squamous epithelium sample), and only in a subset of the samples in the bronchial epithelium. As this is still the most likely route of entry for the virus (nasal epithelial and/or conjunctiva of the eye), it would be great if the authors could discuss the difference between upper (nasal) and lower (bronchi) airways in a bit more detail in the discussion section (which in itself is very well done!). One attractive hypothesis would be that the initial infection (eye/nose) might then cause a classical interferon response which would allow the virus to spread over the respiratory mucosa. The authors of course need not agree with this hypothesis, but it would be great if the difference between nose and bronchi would be made clear (or if the authors disagree with my interpretation, then please indicate so).
- Figure 2 - the three columns to the right, please indicate that all three columns represent ACE2 expression in the figure itself for clarity (the 60/10/1% positivity)

Answers to Reviewer #3

First of all I would like to apologize for the late review - the volume of review requests has increased substantially and unfortunately I was unable to review the resubmission by the indicated time. This is especially unfortunate given the excellent job the authors have done on the revision of this manuscript. It is now a very high-quality manuscript that needs very little if any additional editing. The authors accurately addressed my main concerns and most of the minor concerns with the initial submission, and it's great to see that the manuscript is now very convincing, well-balanced and thorough. In addition, I would like to mention the attractive visualization, which really helps getting the message across.

We thank the reviewer for the inspiring comments and for acknowledging our efforts in this study.

The staining pattern in most barrier tissues is very similar, with a highly apical staining within the epithelium. In all cases, the HPA000288 Ab seems to give a stronger staining of this particular staining pattern than the MAB933. In the respiratory epithelia, this same staining pattern is observed only with the HPA000288 Ab, and is observed in all subjects in the nasal epithelium (with exception of the squamous epithelium sample), and only in a subset of the samples in the bronchial epithelium. As this is still the most likely route of entry for the virus (nasal epithelial and/or conjunctiva of the eye), it would be great if the authors could discuss the difference between upper (nasal) and lower (bronchi) airways in a bit more detail in the discussion section (which in itself is very well done!). One attractive hypothesis would be that the initial infection (eye/nose) might then cause a classical interferon response which would allow the virus to spread over the respiratory mucosa. The authors of course need not agree with this hypothesis, but it would be great if the difference between nose and bronchi would be made clear (or if the authors disagree with my interpretation, then please indicate so).

The staining of ACE2 in nasal mucosa was observed in 50% (6 out of 12) of the individuals with only HPA000288, see Table 1 for details. Only the positive nasal mucosa cases were included in Figure 5.

We thank for the reviewer's perspective in pointing out a recent discovery regarding the dynamics of interferon-mediated effect on ACE2 expression in the respiratory system. Regarding the potential difference in ACE2 expression in upper and lower respiratory cells, we have addressed the matter and added information in the discussion section of the manuscript.

Figure 2 - the three columns to the right, please indicate that all three columns represent ACE2 expression in the figure itself for clarity (the 60/10/1% positivity).

We agree with the reviewer's comment to clarify how the ACE2 expression is being visualized in the figure and have revised Figure 2 accordingly.

Thank you again for sending us your revised manuscript. We are now satisfied with the modifications made and I am pleased to inform you that your paper has been accepted for publication.

Corresponding Author Name: Cecilia Lindskog

Manuscript Number: MSB-20-9610